# A mathematical model of H5N1 influenza transmission in US dairy cattle

**Thomas Rawson** [1] ✉, **Christian Morgenstern** [1], **Edward S. Knock** [1], **Joseph Hicks**[1], **Anh Pham**[1], **Guillaume Morel** [1,2], **Aurelio Cabezas Murillo**[3], **Michael W. Sanderson** [4], **Giovanni Forchini**[1,2], **Richard FitzJohn**[1], **Katharina Hauck** [1] & **Neil Ferguson**[1]

2024 saw a novel outbreak of H5N1 avian influenza in US dairy cattle. Limited surveillance data has made determining the true scale of the epidemic difficult. We present a stochastic metapopulation transmission model that simulates H5N1 influenza transmission through individual dairy cows in 35,974 herds in the continental US. Transmission is enabled through the movement of cattle between herds, as indicated from Interstate Certificates of Veterinary Inspection data. We estimate the rates of under-reporting by state and present the anticipated rates of positivity for cattle tested at the point of exportation over time. We investigate the impact of intervention methods on the underlying epidemiological dynamics, demonstrating that current interventions have had insufficient impact, preventing only a mean 175.2 reported outbreaks. Our model predicts that the majority of the disease burden is, as of January 2025, concentrated within West Coast states. We quantify the uncertainty in the scale of the epidemic, highlighting the most pressing data streams to capture, and which states are expected to see outbreaks emerge next, with Arizona and Wisconsin at greatest risk. Our model suggests that dairy outbreaks will continue to occur in 2025, and that more urgent, farm-focused, biosecurity interventions and targeted surveillance schemes are needed.

In February 2024, dairy farms in Texas, New Mexico, and Kansas began to report an unidentified disease spreading through lactating herds[1,2]. The disease was characterized by decreased rumen activity, diarrhoea, reduced milk production, and thicker milk consistency and discoloration. In March, milk samples from these farms were confirmed via real-time PCR as being infected with highly-pathogenic avian influenza H5N1[3]. This marked the first time that transmission of Influenza A had been identified in US cattle populations[4].

Subsequent phylogenetic studies identified this strain circulating in dairy cattle as a clade 2.3.4.4b genotype first isolated from wild bird populations in late 2023[5]. This, and additional most-recent common ancestor studies, suggests that the initial spillover into cattle likely occurred in December of 2023 in Texas[6]. Histological studies demonstrated the virus' capability to bind to epithelial cells in the mammary tissue of dairy cows[7], in accordance with findings of far greater viral shedding within milk compared to nasal swabs or respiratory tissues[3]. These factors indicate that the repeated use of milking apparatus between individual cows during milking is a primary route of transmission[8,9]. This additionally explains why outbreaks have yet to be detected in beef cattle or dry heifers. In April, the first human spillover case from dairy cattle was reported[10], with a dairy worker demonstrating conjunctivitis but no respiratory

[1]MRC Centre for Global Infectious Disease Analysis, Jameel Institute, School of Public Health, Imperial College London, London, UK. [2]Umeå School of Business, Economics and Statistics, Umeå Universitet, Umeå, Sweden. [3]World Animal Health Information and Analysis Department, World Organisation for Animal Health, Paris, France. [4]Center for Outcomes Research and Epidemiology, College of Veterinary Medicine, Kansas State University, Manhattan, KS, USA. ✉e-mail: t.rawson@imperial.ac.uk

symptoms, likely due to contact with infected milk during the milking process.

The dairy industry is a substantial contributor to US national economic activity, with over 9 million milk cows[11] contributing to approximately 3% of US GDP[12]. Cattle are frequently moved between premises and across states. As a result of this, export of cattle has been implicated in the proliferation of H5N1 to herds nationwide[3], leading to interventions on exports being introduced. When cattle are shipped interstate, they must be accompanied with an Interstate Certificate of Veterinary Inspection (ICVI) to certify that such animals are fit to travel[13,14]. As of April 29th 2024, cattle exported interstate have up to 30 cows in the cohort tested for H5N1 influenza[15]. Should the herd test positive, the export cannot proceed, and the origin herd must be quarantined for 30 days before being tested again. No such requirements were introduced for transfers of cattle within state borders.

As of December 9th 2024, there have been 720 cattle herd outbreaks reported by the USDA[16], across 15 states, and 35 human spillover cases with cattle as the exposure source[17]. Prolonged outbreaks of H5N1 in a novel animal reservoir presents a continuing threat for further spillover and the potential for viral reassortment. Recent structural analysis by Lin et al.[18] suggests that a single glutamine to leucine mutation within this 2.3.4.4b variant would be sufficient to allow for human receptor binding. For this reason, ascertaining the true size of the current epidemic, and identifying the areas of greatest circulation, is crucial to inform public health responses for curbing transmission. In previous bovine disease outbreaks, such as bovine spongiform encephalopathy and foot-and-mouth disease in the UK, public health responses have been significantly aided by modeling studies to estimate rates of under-reporting[19], estimating key epidemiological mechanisms[20], and quantifying the impact of control policies[21]. Such efforts have not yet been applied to the current bovine H5N1 epidemic in the US.

In this study, we estimate the true size of the current epidemic via a stochastic metapopulation transmission model capturing 9,308,707 milk cows distributed across 35,974 herds across the 48 continental US states, as counted in the 2022 agricultural census[11]. Epidemiological parameters are estimated by fitting to outbreak data via a Bayesian evidence synthesis approach[22]. The movement of cattle between herds and states is captured using probabilistic outputs of the US Animal Movement Model (USAMM)[23] and verified using actual 2016 ICVI data[14]. Mechanistic modeling assumptions are made relating the probability of detecting and reporting an infected herd proportional to the number of infected cattle and total population size of the herd, irrespective of the US state they reside in. The model successfully simulates outbreaks for US states that have frequently reported outbreaks, such as California. We estimate the rates of under-reporting by state, by comparing the number of confirmed outbreaks with model simulated trajectories, and present the anticipated rates of positivity for cattle tested upon leaving each state over time. We further use this model to interrogate the impact of intervention methods to date on the underlying epidemiological dynamics, and quantify the extent of uncertainty in the scale of the current epidemic, highlighting the most pressing data streams to capture.

## Results

The model structure and key output metrics are illustrated in Fig. 1. Data on the number of dairy herds in the United States and their respective populations are taken from the 2022 US Agricultural Census[11]. Each herd is modeled via Susceptible-Exposed-Infected-Recovered (SEIR) infection dynamics. Panel 1A illustrates the number of infected cattle per herd over time. Panel 1B depicts the date at which an infected herd probabilistically reports an outbreak. Panel 1C illustrates the aggregated number of herds with any infected cattle per state, and the number of new reported outbreaks. The number of new reported outbreaks is skewed by contact tracing efforts and other

time-varying factors—thus are not independent data samples. Therefore, we do not fit to outbreak incidence data, but rather to the date of first detection of an outbreak in each state (panel 1D).

Figure 2 plots the simulated mean and 95% credible intervals (CrI) of the date of first outbreak detection and the number of reported outbreaks for each US state. After fitting the epidemiological parameters of the model via pMCMC[22,24], we generated 20,000 stochastic realizations of the model with parameter estimates drawn from the posterior distributions of the fit parameters. All model results shown are from these stochastic realizations so as to present the full stochastic range of uncertainty rather than the optimized realizations from the pMCMC fits.

The date of first detection in panel 2A is represented as a step function, where the black line in these plots shows the proportion of simulations that have had their first outbreak reported by that date in the respective state. The shaded areas shows the 95% CrI of the modeled date of first outbreak in each state. Note that for the majority of states in panel 2A, such as Washington, the upper 95% CrI bound is the final date of the simulations. This should not be interpreted as dates beyond this point therefore lying outside of the 95% CrI.

Panel 2B shows the proportion of dairy herds in each state reporting new outbreaks each week from December 18th 2023 to December 2nd 2024. Both panels illustrate that the majority of outbreaks are currently concentrated along the West Coast of the country. The model forecasts that states in the mid-West and Florida are the most probable next states to declare their first outbreak. This trend is due to the epidemic beginning in Texas, which exports primarily to nearby West Coast states.

The model is seen to overestimate the number of reported outbreaks in some states. For example, Texas, New Mexico, and Ohio all feature simulations whose credible interval does not contain the observed data. While our model assumes differences in outbreak detection due to differences in herd sizes by state, we do not assume further intrinsic state-varying differences in outbreak detection. In reality, differences in public health resourcing and messaging will impact outbreak detection rates. 72% of outbreaks reported as of December 9th 2024 have been in California. Due to making up the majority of the epidemiological data, model fits are mostly tuned to the detection rates observed in California. Therefore, overestimation of the model can be interpreted as under-reporting within a state compared broadly to baseline reporting efforts in California, as seen most strongly in the case of Arizona (Fig. 2A). The simulated number of infected herds, the number of herds with any infected cows on the premises, is shown in Supplementary Material Section 3.1.

Twenty-six of the 48 US states (54%) observed an outbreak of H5N1 before December 2nd 2024 in the majority of model simulations (> 50% of simulations, Table 1). Based on these probabilities, one would expect to have observed outbreaks in a mean of 27 (22–32 95% CrI) states by December 2nd 2024, assuming all states reported outbreaks equally. In actuality, only 16 states identified and reported outbreaks in this time period, indicating a high degree of under-reporting compared to the high baseline set by California.

We note that simulated incidence levels have a bimodal distribution. Many simulations never see H5N1 emerge in a particular state, which is why the 95% CrIs in Fig. 2 often span 0. Thus, this mean value is not the most probable outcome, but should be interpreted alongside the proportion of simulations which see no infections in particular states, as provided in Table 1. Particularly narrow 95% CrIs are seen in Fig. 2A for Texas, Ohio, New Mexico, and Kansas, due to the seeding of cases in these states as detailed in the Methods.

These results demonstrate how the composition of the dairy sector in each state has a significant impact on the overall epidemic dynamics. For example, while Florida is increasingly likely to report an outbreak (Fig. 2A), the expected proportion of herds reporting outbreaks in Florida remains low (Fig. 2B). First, states with larger herd

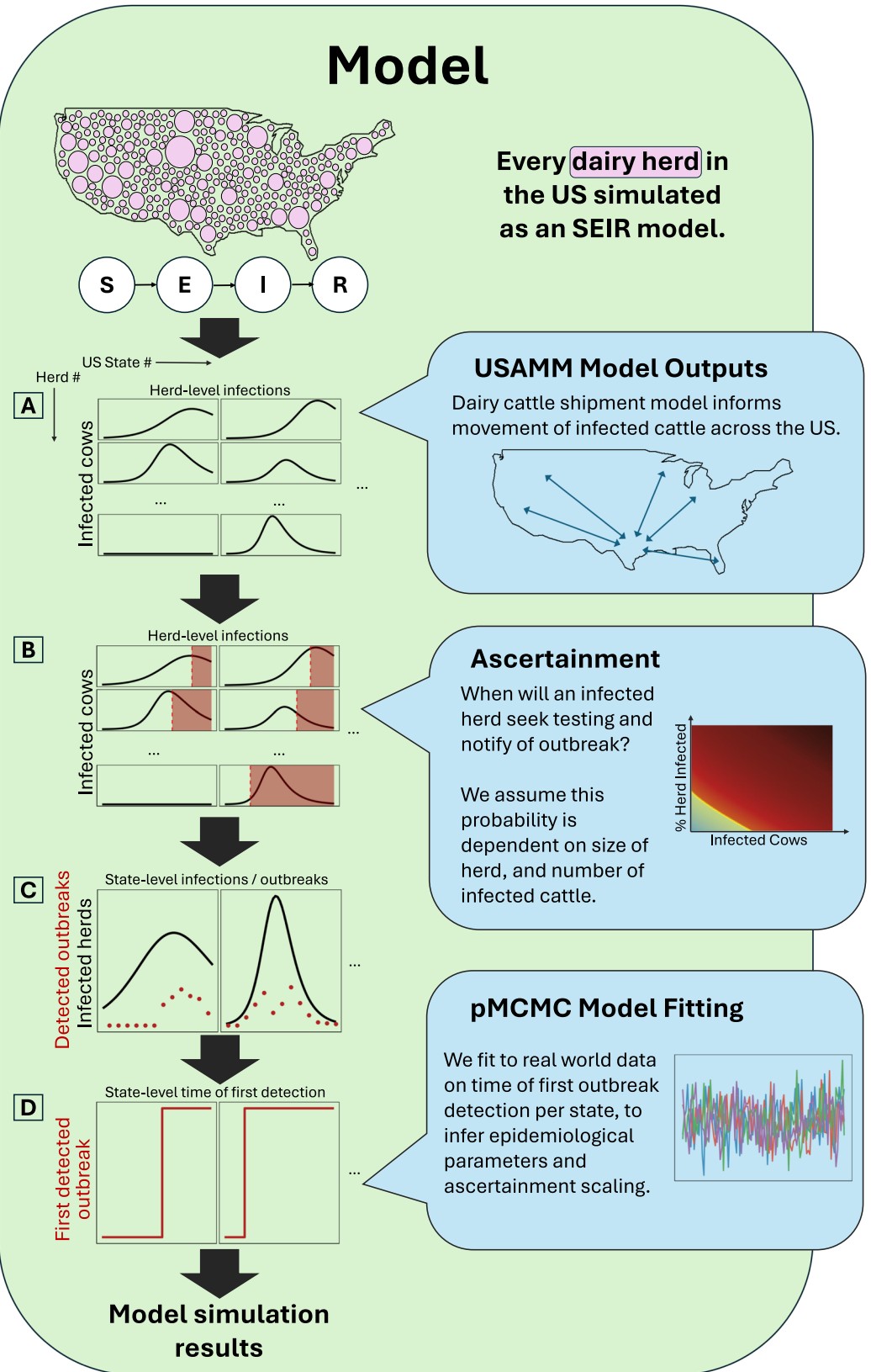

**Fig. 1 | Schematic overview of model format and outputs.** Infection spreads from the initial infected state through export of cattle. **A** Cattle exports are stochastically generated using trade data from the United States Animal Movement Model (USAMM)[23]. **B** At each time step, a herd has a probability of testing, and notifying of an outbreak. **C** We aggregate the number of herds with any infected cattle by state, and the number of newly reported outbreaks, at each date. **D** We fit global epidemiological parameters and an ascertainment scaling parameter via particle Markov Chain Monte Carlo simulation (pMCMC). Using the posterior distributions of these parameters, we are able to produce further model simulations herein. Full methodological details are presented in Supplementary Material Section 2.

**A** Date of First Outbreak Detection

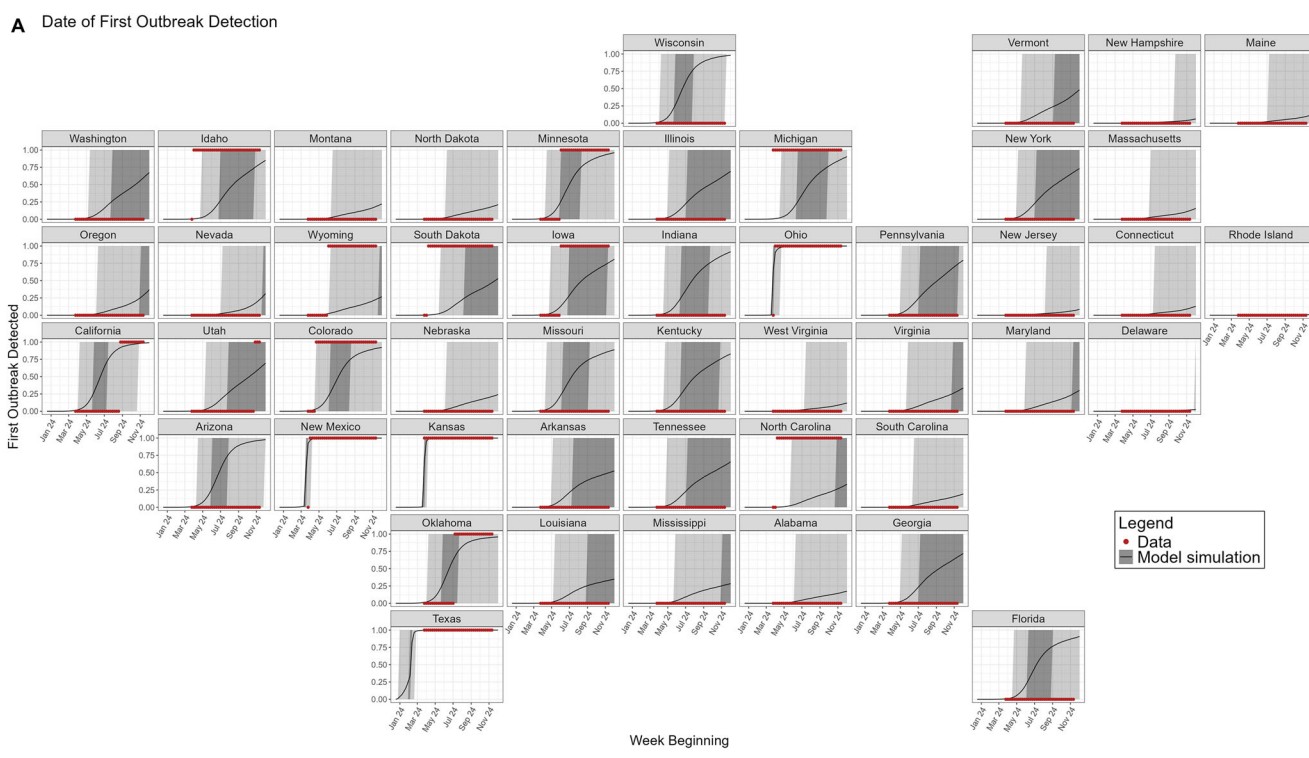

**B** Proportion of Herds Declaring Outbreaks Weekly

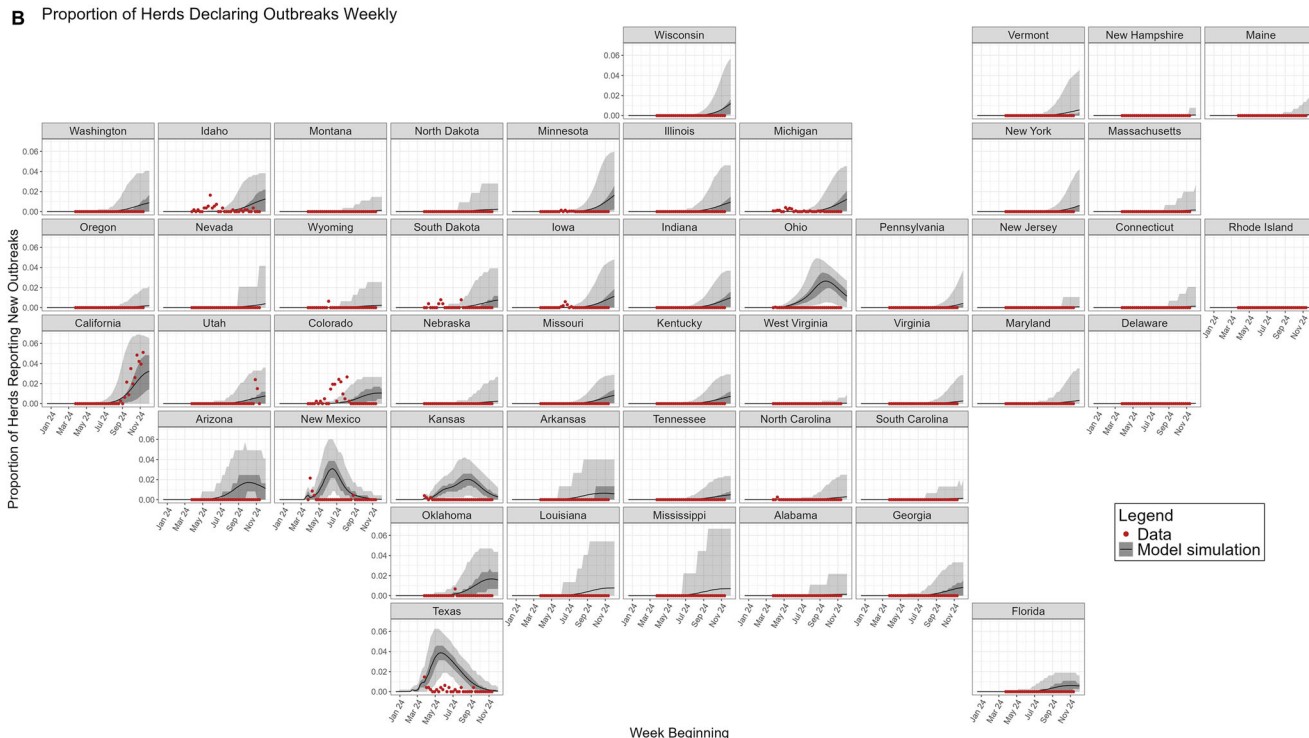

**Fig. 2 | Model simulations.** After fitting model parameters we simulate 20,000 stochastic realizations drawing from the parameter posterior distributions. Displayed is the epidemic trajectory from these simulations for each US state. **A** shows the date at which the first outbreak is detected in a state, a binary outcome. 0 indicates the state has not yet reported its first outbreak. 1 indicates that it has. Model simulation thus plots the proportion of the 20,000 realizations which have simulated a reported outbreak by this date. **B** shows the proportion of herds in each state which report new outbreaks per week, assuming no differences in ascertainment (parameter $A^{asc}$) between states. Red points depict data. The black line depicts the model mean, lightly shaded grey region depicts the 95% credible interval (95% CrI), and the darker shaded grey region depicts the 50% CrI.

**Table 1 | Reported outbreaks**

| US State | Up to and including the week beginning December 2nd 2024 | | |
|---|---|---|---|
| | Outbreaks reported (Observed) | Simulation outbreaks reported Mean (95% CrI) | Probability of no outbreaks (Proportion of Simulations) |
| California | 520 | 339 (3–809) | 0.010 |
| Colorado | 64 | 57 (0–139) | 0.077 |
| Idaho | 35 | 64 (0–256) | 0.150 |
| Michigan | 29 | 136 (0–710) | 0.096 |
| Texas | 26 | 322 (197–376) | 0.000 |
| Iowa | 13 | 89 (0–512) | 0.191 |
| Utah | 13 | 25 (0–133) | 0.306 |
| Minnesota | 9 | 249 (0–1305) | 0.039 |
| New Mexico | 9 | 86 (74–97) | 0.000 |
| South Dakota | 7 | 19 (0–119) | 0.471 |
| Kansas | 4 | 194 (74–279) | 0.000 |
| Oklahoma | 2 | 70 (0–158) | 0.041 |
| Nevada | 1 | 2 (0–17) | 0.686 |
| North Carolina | 1 | 11 (0–113) | 0.667 |
| Ohio | 1 | 1004 (279–1487) | 0.000 |
| Wyoming | 1 | 5 (0–48) | 0.733 |
| Alabama | 0 | 2 (0–25) | 0.825 |
| Arizona | 0 | 34 (1–51) | 0.023 |
| Arkansas | 0 | 8 (0–34) | 0.476 |
| Connecticut | 0 | 2 (0–36) | 0.870 |
| Delaware | 0 | 0 (0–1) | 0.974 |
| Florida | 0 | 35 (0–78) | 0.094 |
| Georgia | 0 | 33 (0–155) | 0.282 |
| Illinois | 0 | 48 (0–316) | 0.309 |
| Indiana | 0 | 119 (0–598) | 0.083 |
| Kentucky | 0 | 69 (0–362) | 0.171 |
| Louisiana | 0 | 8 (0–56) | 0.652 |
| Maine | 0 | 3 (0–45) | 0.879 |
| Maryland | 0 | 9 (0–115) | 0.698 |
| Massachusetts | 0 | 3 (0–40) | 0.843 |
| Mississippi | 0 | 5 (0–37) | 0.716 |
| Missouri | 0 | 125 (0–562) | 0.112 |
| Montana | 0 | 4 (0–48) | 0.780 |
| Nebraska | 0 | 7 (0–80) | 0.759 |
| New Hampshire | 0 | 1 (0–5) | 0.938 |
| New Jersey | 0 | 1 (0–13) | 0.914 |
| New York | 0 | 108 (0–882) | 0.268 |
| North Dakota | 0 | 3 (0–37) | 0.790 |
| Oregon | 0 | 9 (0–104) | 0.631 |
| Pennsylvania | 0 | 103 (0–888) | 0.205 |
| Rhode Island | 0 | 0 (0–0) | 0.990 |
| South Carolina | 0 | 3 (0–33) | 0.808 |
| Tennessee | 0 | 34 (0–199) | 0.343 |
| Vermont | 0 | 24 (0–230) | 0.516 |
| Virginia | 0 | 16 (0–185) | 0.664 |
| Washington | 0 | 33 (0–193) | 0.326 |
| West Virginia | 0 | 2 (0–22) | 0.881 |
| Wisconsin | 0 | 454 (1–2729) | 0.019 |

For each US state we present the observed number of reported outbreaks, and the number of reported outbreaks predicted by our model. Mean and 95% CrIs are provided from 20,000 stochastic realizations. We also display the proportion of these simulations for which no outbreaks were reported in each state.

sizes present greater opportunities for infection to spread quickly within the respective holdings. This then poses a greater risk of contaminating neighboring herds through shared workers, equipment, grazing space, or environmental runoff. Secondly, larger population holdings are observed to import larger numbers of cattle, hence increasing the probability of infection, as only up to 30 cows are currently tested during inter-state transfer[15]. Thirdly, our model assumptions of ascertainment trend towards larger holdings being more likely to report outbreaks, as has been observed in real-world reporting to date[3] (Fig. 3). The respective sizes of each state's dairy industry is provided in Supplementary Material Section 1.

Our model assumes each herd that has not yet reported an outbreak, has a probability of declaring an outbreak at each date. This probability is dependent on the absolute number of infected cattle in the herd, and the proportion of the herd that is currently infected. This functional form (Fig. 3A) was designed after discussion with veterinarians based on their experience with on-farm callouts. This baseline probability is then further scaled by an ascertainment rate model parameter, which is estimated in model fitting (Table 2). Alternate ascertainment rate assumptions are presented as sensitivity analyses in section 3.2.3 of the Supplementary Material.

We calculate the mean probability that a randomly selected herd in each state will report an outbreak, given that 10% of its animals are infected. These values ranged from 0.412 in California, a state with a greater number of large herds, to 0.092 in West Virginia (Fig. 3B, C). We see that states with a greater number of large herds are more likely to report outbreaks than other states. Correspondingly, California has reported the vast majority of outbreaks to date (Table 1).

Current federal orders require that, when exporting cattle interstate, up to 30 randomly-chosen cows from the exported cohort will be tested for H5N1, and only if all tested cattle register negative tests will the export take place[15]. Thus, exports of less than 30 cattle will have all cows tested, and exports of more than 30 cattle will have only 30 randomly selected cows tested. The results of these tests, be it positive or negative, are not currently reported to health authorities. We output from our model simulations the expected rates of export test positivity per state. This takes into account the expected number of cattle being exported.

Figure 4 shows the mean probability by state of such an export testing positive. We use the 20,000 simulation runs produced in Fig. 2 to sample 20,000 national epidemic trajectories for each herd. For each herd, and for each time point, we assume that it exports cattle, and sample how many cattle it will be exporting. We then calculate the probability of these cattle testing positive via the density of a hypergeometric distribution. Figure 4 displays the mean probability over all herds and all 20,000 stochastic realizations. The 95% CrIs are provided in Supplementary Material Section 3.1.

Lastly, we use the model to assess the impact that interstate testing has had on the epidemic trajectory. We consider two counterfactual scenarios. Scenario 1) weaker measures—we assume no restrictions are introduced, no testing is required when exporting cattle, and thus all interstate exports proceed unabated. Scenario 2) stronger measures—we assume that the federal order was implemented 28 days earlier, on April 1st 2024, and that up to 100 cattle are tested instead of 30.

Considerable stochastic variation is seen across all scenarios, though we do see a reduction in all infection measures for the mean values of scenario 2—stronger measures, and an increase for the mean values of scenario 1—weaker measures, compared with the baseline scenario (Fig. 5). For the week beginning December 2nd 2024, under baseline model assumptions, the model simulates a national total of mean 120.9 new reported outbreaks (15–518 95% CrI), compared to an increased mean of 150.7 outbreaks (95% range 17–632 under the no interventions scenario 1, and a reduced mean of 93.4 outbreaks (95% range of 11–407) under the stronger measures of scenario 2.

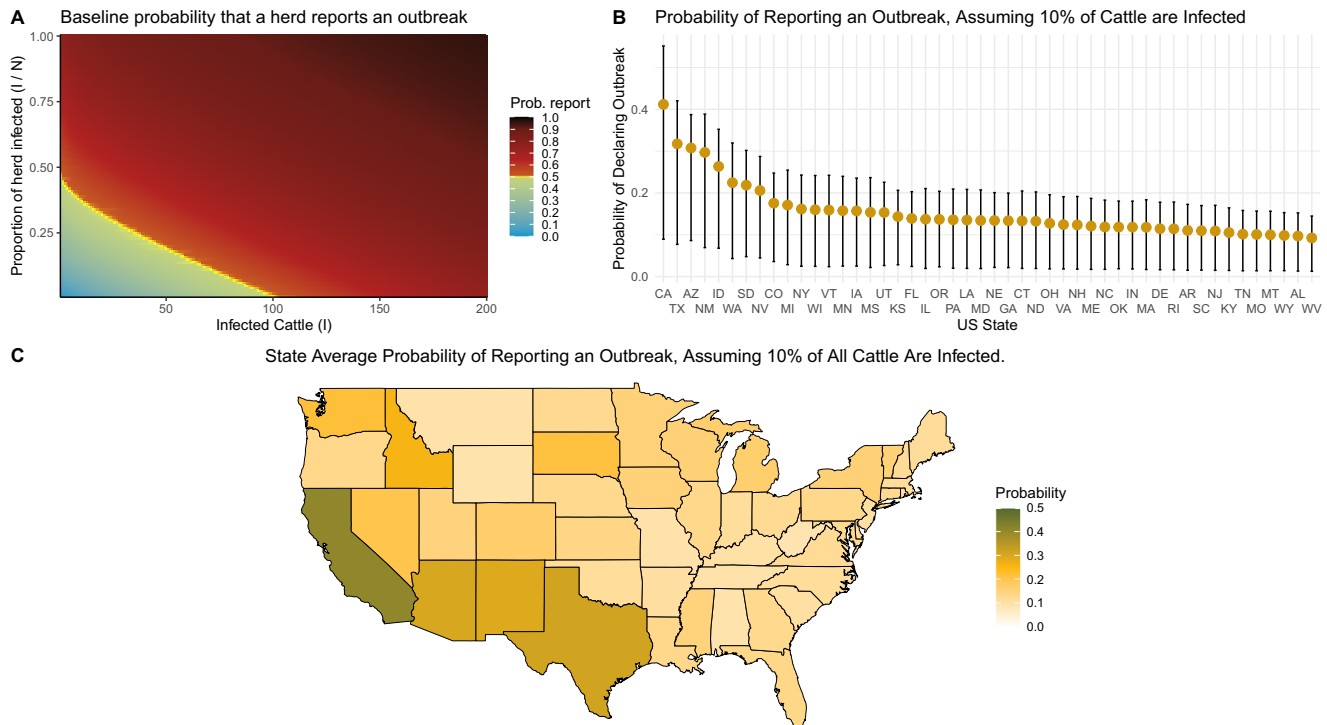

**Fig. 3 | Ascertainment rate assumptions. A** shows how the modeled baseline probability of reporting an outbreak depends on the number and proportion of infected cattle in a herd. Our model assumes that the probability that an infected herd reports an outbreak depends on the size of the holding, and the number of infected cattle on that date. **B** shows the mean and 95% CrI per-herd probability a herd reports an outbreak by US state, assuming every herd has 10% of its cattle infected. The credible interval captures the variation in herd sizes and the posterior distribution of the ascertainment rate parameter. **C** maps the mean values shown in (**B**).

**Table 2 | The Prior distributions and posterior intervals for all fit model parameters**

| Parameter | Description | Prior distribution | Posterior–Median (95% Crl) |
|---|---|---|---|
| $\frac{\beta}{\gamma}$ | transmission rate / recovery rate | Uniform (0.05, 3) | 1.864 (0.929–2.932) |
| $\alpha$ | Intra-state transmission coefficient | Uniform (0, 0.1) | 0.063 (0.009–0.098) |
| $\sigma$ | Incubation rate | Uniform (0.05, 2) | 1.050 (0.199–1.956) |
| $\gamma$ | Recovery rate | Uniform (0.05, 2) | 1.084 (0.384–1.942) |
| $A^{asc}$ | Ascertainment rate scaling | Beta (1, 1) | 0.648 (0.091–0.986) |

Figure 5 shows that under each scenario, the epidemic continues to grow—meaning border testing measures alone are insufficient to effectively curb the epidemic. Stronger, farm-focused intervention measures would be required to reduce transmission sufficiently to achieve control.

### Sensitivity analyses

All results are also produced under four alternate modeling assumptions. Supplementary Material section 3.2.1 considers alternate likelihood assumptions. Supplementary Material section 3.2.2 infers cattle exports from exact 2016 ICVI export data. Supplementary Material section 3.2.3 considers simplified ascertainment rate assumptions—where ascertainment is proportional only to the proportion of the herd infected. Due to the relatively short time frame considered, and unclear evidence as to the extent of mortality or culling, we did not include birth-death processes within our model. Supplementary Material section 3.2.4 considers the dynamic impact of including such birth-death mechanisms. Our conclusions are unchanged in all of these sensitivity analyses.

## Discussion

Our study presents the first herd-level dynamic model of highly pathogenic avian H5N1 influenza transmission in US dairy cattle across the continental United States. By synthesizing existing data on dairy herd population sizes and cattle trade patterns, we recreate the spread of the virus from an initial seeding in Texas on December 18th 2023, through to the week beginning December 2nd 2024.

The model projects that the majority of the initial national disease burden is focused within West Coast states, due to their existing trade patterns with Texas, and the size of their respective dairy industries. However, East Coast states are not without risk of currently housing infected herds, as our model suggests that a considerable degree of under-reporting is misrepresenting the true size of the epidemic. A clear result from Fig. 2 and Table 1 is that some states are particularly likely to be home to infected herds, but have yet to identify and report infections. Most notable are Arizona, Wisconsin, Indiana, and Florida. Arizona has the largest mean herd size in the country (Supplementary Material Section 1), and extensive trade connections with Texas and California (Supplementary Material Section 2.4)—states particularly burdened with infection. Wisconsin, while farther from the epidemic epicenter, has the largest number of dairy herds in the country—6216. While Florida has a modestly sized dairy sector, and is located on the east coast, it has one of the highest mean herd sizes in the country, as their industry is predominantly made up of a few very large holdings. It also imports more cattle from Texas than its neighbors. Indiana

presents itself as having a high likelihood of probable infection due both to having a very high number of dairy herds, but also due to its frequent trading links with Wisconsin. Table 1 shows that, while it is not implausible that no infections have established within these states, the probability of this is low, with Wisconsin in particular only reporting no outbreaks in 1.9% of model simulations. In only 22 of the 48 continental US states did our model predict zero reported outbreaks in > 50% of model simulations (Table 1). Figure S20 of the Supplementary Material visualizes the herd population sizes of each state against the frequency of imports from Texas, demonstrating the relationship between herd sizes and outbreak likelihood.

The model also demonstrates how the distribution of cattle populations in each state mechanistically impacts the rate of reporting. Figure 3 shows that, due to many West Coast states housing large populations of dairy cattle in single herds, they have a higher-than-average likelihood of reporting outbreaks. This is reflected in the outbreak data. California has reported over 8 times as many outbreaks as the state with the next highest number of reported outbreaks. Our model suggests that this can be explained by the fact that the average herd size in California is significantly higher, and not necessarily due to more robust epidemiological investigation attempts in the state.

The only national intervention mandated to date is the testing of cattle exported interstate. Up to 30 cows in an exported cohort are tested for H5N1, and must test negative for the export to proceed. Figure 4A shows that, early in the epidemic, Texas was one of the only states with a non-negligible probability of cattle testing positive at export, though we note that such interventions were only brought in from April 29th 2024. By August (panel 4B), Texas had a greater than 40% mean probability of an export testing positive. By December of 2024, our model predicts that infections in Texas may have begun to decrease, and a more uniform probability of positivity is observed across the country. According to the USAMM, a mean 29,590 (IQR 922) interstate exports of dairy cattle occur every year[23]. Given that such testing is mandated to occur, it would be prudent to report such testing to verify against our expected positivity rates and better refine model estimates.

Our model has also demonstrated that the border-testing intervention alone, while a valuable (if unrealised) opportunity for surveillance, is insufficient to control the spread of H5N1 influenza. We explored the counterfactual scenario of stronger border testing measures, of up to 100 cows, and introduced 28 days earlier, on April 1st 2024. Despite a slight reduction in the mean number of outbreaks under this scenario, the fundamental epidemic dynamics remained unchanged, with infections and outbreaks continuing to increase as the year continued. This suggests that targeted biosecurity interventions at farm level, such as postmilking teat dipping and the use of disposable wipes for premilking teat disinfection[25], and interventions between herds such as boot dips at facility entrances, clothing disinfection post-site visit, or greater emphasis on adequate personal protective equipment[26] will be required (Supplementary Fig. S19). Additionally, better outreach with industrial partners should be pursued. On May 10th 2024, the U.S. Department of Agriculture (USDA) provided a total of $98 million to support biosecurity measures[27,28], whereby individual farms could apply for up to $28,000 to implement protocols such as secure milk plans, disposal of infected milk, veterinarian costs, and testing costs. As of January 9th 2025, only 510 premises have applied for this additional funding[29]. On May 30th 2024, the USDA announced a further $824 million was being allocated to a nationwide voluntary Dairy Herd Status Pilot Program, whereby premises could apply for free routine milk surveillance. The 2022 US Agricultural Census lists 36,024 dairy farms. As of January 9th 2025, only 75 herds have enrolled for the voluntary testing program[30]. Evidently, voluntary measures are currently failing to see sufficient uptake.

Data availability has been poor throughout the epidemic, the only epidemiological data stream being the number of reported outbreaks. Due to a lack of uniform surveillance or testing, uncertainty surrounding state-level infection levels is large, as demonstrated in Fig. 2. Uncertainty is further compounded by the probabilistic nature of our modeled export assumptions, necessitated by a lack of precise movement data in this period. Many other countries, including the European Union, enforce mandatory identification of all premises, individual cattle, and movement of animals, often by electronic tagging methods[31]. The US has no such requirement. Additionally, since veterinary and public health responses are governed at the state level, individual states vary greatly in the measures, resources, and interventions they have applied to limit spread. Reported outbreak incidence data are not sufficient to reasonably quantify these state-level differences. The most valuable enhancement to current surveillance would be through stratified and systematic sentinel testing for infection, reporting of both positive and negative test results. This would allow overall assessment of infection prevalence within farms, and estimation of the proportion of herds with any level of infections, which in turn would allow better estimation of the risks of onward infection through cattle trade. A further additional valuable source of data would be the publication of the results of pre-export cattle testing currently being undertaken. Figure 4 shows our estimates of the rates of positive tests at export currently, which such data might be compared against, if released.

While our analysis suggests that some of the earliest infected states may have passed the peak of their epidemics, Fig. 2 suggests that many more states will still be in the early stages of their epidemics. Importantly, our model also does not capture the role of either re-infection, or the emergence of new, more adapted, clades of the virus (though studies have shown that initial infection infers strong protection against reinfection[32]). Our analysis suggests that dairy herd outbreaks will continue to be a significant public health challenge in 2025, and that more urgent interventions are sorely needed. Early economic models of the impact of the epidemic on the US dairy sector project economic losses ranging from $14 billion to $164 billion[12]. Additionally, 35 human spillover cases from cattle[17] have been reported to date. The longer the epidemic persists in a novel mammalian reservoir, the greater the risk of further human spillovers and viral adaptations to human hosts. Recent research suggests only minimal genetic distance separates the currently circulating clade from adaptation to human receptor binding[18], and such adaptation has already occurred to improve virus replication in bovine and primary human airway cells[33].

Our work is not without limitations. Most importantly is that, due to insufficient epidemiological data, we had to make strong assumptions about the probability of ascertainment—whether or not an infected herd is identified and reported. Figure 3 outlines the implications of these assumptions, but the wide credible interval for our estimate of the ascertainment parameter $A^{asc}$ reflects these data limitations. Additionally, because the US does not employ a mandatory electronic tagging system, there is no way to accurately capture the precise cattle movements for 2024. While we were provided with the 2016 ICVI data utilised in Cabezas et al.[14], it was considered, upon comparison with USAMM model simulations, that precise inter-state exports might vary greatly year-to-year. Therefore, assuming identical movements to 2016 could induce significant bias into the results. Thus, we instead take the probabilistic approach, whereby the exports of cattle are probabilistically determined through model simulations according to the USAMM model[23]. While this introduces further uncertainty into the model, it accurately demonstrates how poor data availability regarding precise 2024 cattle movement hampers epidemic forecasting efforts. We nonetheless present model results fit using this 2016 ICVI data as a sensitivity analysis in Supplementary Material Section 3.2.2.

Additionally, our work does not consider the dynamic impact of other zoonotic reservoirs. The ongoing H5N1 epidemic in the US is also heavily impacting the poultry industry, with 662 counties reporting

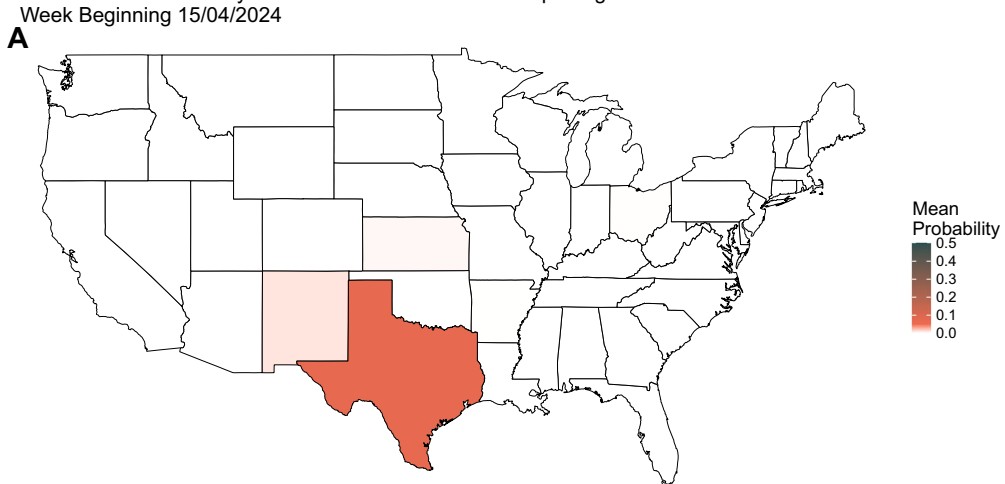

Mean Per−Herd Probability of H5N1 Positive Test When Exporting From State
Week Beginning 15/04/2024

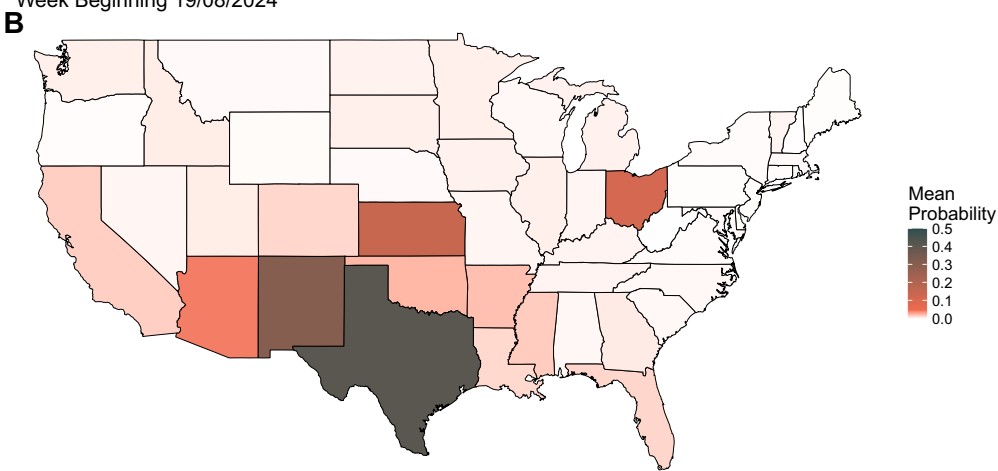

Mean Per−Herd Probability of H5N1 Positive Test When Exporting From State
Week Beginning 19/08/2024

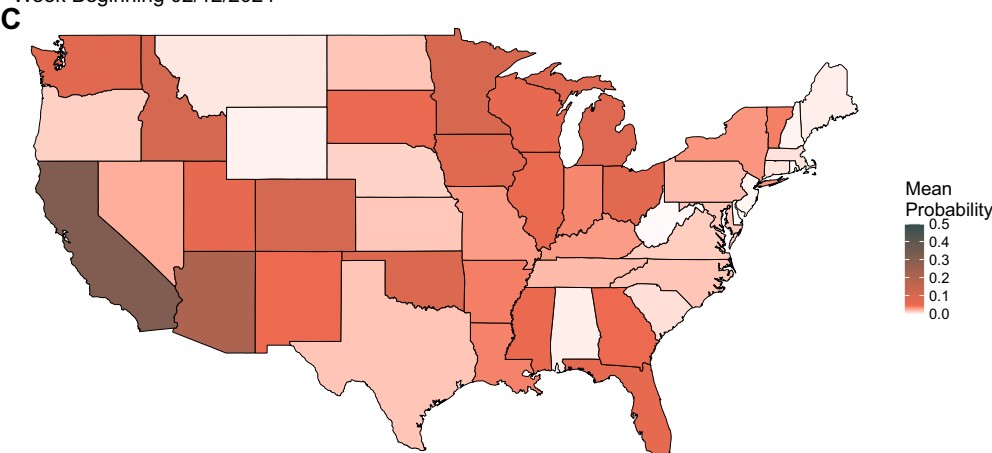

Mean Per−Herd Probability of H5N1 Positive Test When Exporting From State
Week Beginning 02/12/2024

**Fig. 4 | Probability of positive border testing.** We calculate the probability of an export of cattle out of each state testing positive from 20,000 stochastic model simulations. When moving cattle inter-state, up to 30 cattle will be tested for H5N1 per export. Panels show the state average per-herd probability that, should a herd export cattle, it would test positive at: **A** week beginning April 15th 2024, **B** week beginning August 19th 2024, and **C** week beginning December 2nd 2024.

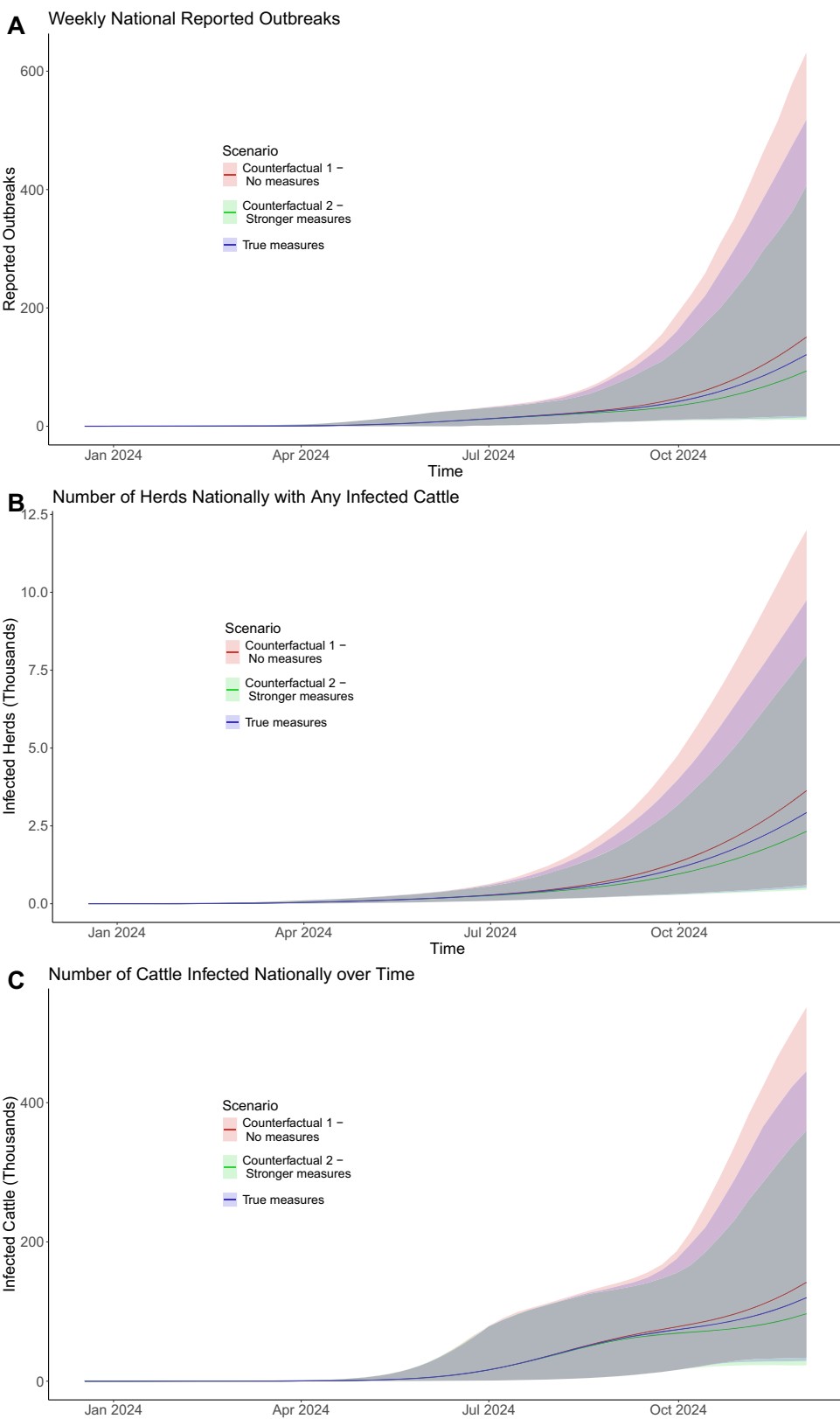

**Fig. 5 | Border testing intervention counterfactuals. A** The number of new reported outbreaks weekly. **B** The number of herds nationally with any infected cattle. **C** The total number of infected cows nationally over time. Solid lines show simulation mean. Shaded regions show 95% CrI. Blue (True measures) depicts baseline model assumptions, whereby up to 30 cows in each inter-state export are tested starting from April 29th 2024. Red depicts the scenario with no border testing. Green depicts border testing of up to 100 cows from each export, implemented 28 days earlier, on April 1st 2024.

outbreaks as of March 3rd 2025[34]. Modeling the disease in poultry is significantly more challenging due to the role played by wild bird migration[35], and our current model does not consider spillover from other animal populations. Further work identifying farm sites which house multiple host species would be an important next step in identifying points of spillover risk between reservoir animals, presenting a risk of further genetic reassortment.

In conclusion, our model demonstrates that we cannot definitively conclude that the current number of reported outbreaks is a true representation of the scale of the current H5N1 influenza epidemic in dairy cattle. Significant under-reporting is likely, and the differences in dairy herd population distributions across states have aided in spreading disease across the west coast. Current mandatory interventions are insufficient for controlling the spread of disease, and voluntary testing and interventions are severely under-utilised. Significant increases in testing are urgently required to reduce the uncertainty of model projections and provide decision-makers with a more accurate picture of the true scale of the national epidemic.

## Methods

### Infection seeding

We seeded the epidemic with five infected cows in a mid-size herd in Texas, on the week beginning December 18th 2023, based on phylogenetic analyses[6]. For the stochastic realizations, we also seeded 9 additional herds in accordance with the nine early outbreaks detailed in Caserta et al.[3]. The herd size, number of infected cattle, and date of seeding is consistent with the data presented in that manuscript.

### Epidemiological dynamics

We construct a stochastic metapopulation SEIR model[36] with 35,974 individual herds of varying population size, informed by the 2022 US Agricultural Census[11]. Each herd's infection dynamics are the stochastic equivalent of the following set of ordinary differential equations (ODEs):

$$
\begin{aligned}
\frac{dS_i^s}{dt} &= -\beta S_i \left( \frac{I_i}{N_i} + \alpha \frac{I_{-i}}{N_{-i}} \right), \\
\frac{dE_i}{dt} &= \beta S_i \left( \frac{I_i}{N_i} + \alpha \frac{I_{-i}}{N_{-i}} \right) - \sigma E_i, \\
\frac{dI_i}{dt} &= \sigma E_i - \gamma I_i, \\
\frac{dR_i}{dt} &= \gamma I_i.
\end{aligned} \tag{1}
$$

Here, $S_i$, $E_i$, $I_i$, and $R_i$ are the number of susceptible, exposed, infected and recovered cows in herd $i$. $N_i$ is the total population of herd $i$. $\beta$, $\sigma$, and $\gamma$ are the transmission, incubation, and recovery rates respectively. $\alpha$ is a model parameter between 0 and 1 controlling the rate of transmission between herds in the same state. $I_{-i}$ and $N_{-i}$ are the total number of infected cattle, and the total number of all cattle, in the US state herd $i$ resides in, not including the cattle in herd $i$ itself. Early epidemiological surveys of farms reporting outbreaks found that transmission routes existed between herds in the same state through the shared use of equipment, staff, or the movements of wild birds[37], which we capture here in the model. We assume no such forms of transmission can occur between herds in different US states.

The stochastic analogue of the above ODEs, is that we calculate the number of cattle progressing between epidemiological compartments via binomial distributions, for each time step $dt$ as:

$$
\begin{aligned}
n_{SE}^i &\sim \text{Binomial} \left( S_i, 1 - \exp \left( -\beta \left( \frac{I_i}{N_i} + \alpha \frac{I_{-i}}{N_{-i}} \right) dt \right) \right), \\
n_{EI}^i &\sim \text{Binomial} \left( E_i, 1 - \exp(-\sigma \, dt) \right), \\
n_{IR}^i &\sim \text{Binomial} \left( I_i, 1 - \exp(-\gamma \, dt) \right).
\end{aligned} \tag{2}
$$

Here $n_{XY}^i$ is the number of cattle moved from compartment $X$ to $Y$ (for general $X$ and $Y$), in herd $i$, in a time step of size $dt$.

After all cattle movements between epidemiological compartments is concluded, we calculate for each herd that has yet to report an outbreak, whether or not it will report an outbreak in that time step. It reports an outbreak with probability $P_i^{\text{outbreak}} = 1 - e^{-\phi_i}$, where $\phi_i$ is

$$
\phi_i = \left( \frac{I_i}{(0.7 N_i)^{0.95}} + \frac{I_i}{150} \right) A^{\text{asc}} \, dt, \tag{3}
$$

and $A^{\text{asc}}$ is a model parameter that we fit. The bracketed term to the left of $A^{\text{asc}}$ in Eq. (3) is shown in the heatmap of Fig. 3A. This functional form was developed in consultation with veterinarians based on their experiences of at what stage of pathogen spread they are typically consulted. While US states undoubtedly vary in their detection capabilities, there is insufficient outbreak data to fit unique $A^{\text{asc}}$ values for each state. Assuming one national $A^{\text{asc}}$ parameter allows us to identify which states that have reported 0 outbreaks to date are driven mostly by under-reporting (Fig. 2B).

### Movement of cattle between herds

After calculating the movement between epidemiological compartments and any reporting of outbreaks, we then calculate the movement of cattle between herds. As detailed in Supplementary Material Section 2.4, we infer from the USAMM the probability, $P_k^{\text{export}}$, for each US state, $k$, that a herd within that state will export cattle each week. We assume the same probability for every herd in the state. We also calculate the proportion of cows in the origin herd that will be exported—$P_k^{\text{export size}}$ from the USAMM export simulations, which include cohort size and size of origin herd. We also calculate the probabilities of, should an export of cattle occur, which US state they will be exported to. This is parameterized by a movement matrix $M$, where element $M_{k,l}$ denotes the probability that an export from state $k$ will go to state $l$. This matrix describes the patterns of interstate movement, and the diagonal represents the probability of an export remaining within the same state. The exact matrix is provided as Supplementary Data. Once the destination state is determined, we randomly allocate which herd in the destination state the cattle will be exported to, scaled by the population size of the respective herds, to preserve herd sizes. Once an origin herd, $i$, and destination herd, $j$, are assigned, we draw the number of cattle to be exported as

$$
\begin{aligned}
n_{S_i S_j} &\sim \text{Binomial} \left( S_i, P_k^{\text{export size}} \, dt \right), \\
n_{E_i E_j} &\sim \text{Binomial} \left( E_i, P_k^{\text{export size}} \, dt \right), \\
n_{I_i I_j} &\sim \text{Binomial} \left( I_i, P_k^{\text{export size}} \, dt \right), \\
n_{R_i R_j} &\sim \text{Binomial} \left( R_i, P_k^{\text{export size}} \, dt \right),
\end{aligned} \tag{4}
$$

where $k$ is the US state that origin herd $i$ resides in. Lastly, before moving cattle between the respective compartments of herds $i$ and $j$, we simulate the border testing mandate. If the model date is after April 29th 2024, we draw a random variable, $X$ from a hypergeometric distribution:

$$
X \sim \text{Hypergeometric} \left( n_{I_i I_j}, \, n_{S_i S_j} + n_{E_i E_j} + n_{R_i R_j}, \, \min(30, n_{N_i N_j}) \right). \tag{5}
$$

Here the three parameters of the above hypergeometric are, the number of success items in the population, the number of failure items in the population, and the number of samples taken without replacement from the population. $X$ is the number of infected cattle drawn. If $X = 0$, then no infected cattle are detected, and the export takes

place. Note, a positive test prevents the export, but does not immediately register as a reported outbreak. All probabilities and a full logic flow diagram are presented in Supplementary Material Section 2. U.S. state boundaries were obtained using the maps package in R (via `map_data("state")`) and visualized with `ggplot2`.

### cowflu **package**

To efficiently simulate the above probabilistic model, we produced a custom *R* package, `cowflu`[38], which allows simulating and fitting the model via the dust2 package[22] in *R*, while the model itself is written in C ++. Documentation on the use of the package and worked vignettes can be found on our github repo: https://github.com/mrc-ide/cowflu. The package is flexible to being applied to any SEIR metapopulation model with custom probabilities of movement between sub-populations, subject to user-defined movement matrices.

### Model fitting

Five of the above model parameters—$\beta$, $\alpha$, $\sigma$, $\gamma$, and $A^{\mathrm{asc}}$, are fit via particle Markov Chain Monte Carlo[24] methods. We assign weakly-informative prior distributions, informed by early studies associated with the current outbreak[39]. We fit the model simulated values of date of first outbreak detection (as seen in Fig. 2A) to the real world data equivalent, via a likelihood function detailed in Supplementary Material section 2.5. We ran the pMCMC simulations across 16 chains of 40,000 iterations each. Model convergence statistics are presented in Supplementary Material section 2.5.

Table 2 shows the priors and posteriors for all model parameters. Note that we fit $\frac{\beta}{\gamma}$ instead of $\beta$ due to observed correlation between $\beta$ and $\gamma$, so as to improve chain mixing.

### Reporting summary

Further information on research design is available in the Nature Portfolio Reporting Summary linked to this article.

## Data availability

All model code and data is available in our associated Github repo: https://github.com/mrc-ide/cowflu.

## Code availability

All model code and data is available in our associated Github repo: https://github.com/mrc-ide/cowflu. A DOI-linked release for this publication is provided at: 10.5281/zenodo.15228688[38]. Additional analysis was performed using the following R packages: `coda` v0.19-4.1, `dplyr` v1.1.4, `lubridate` v1.9.3, `tidyr` v1.3.1.

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

## Acknowledgements

T.R., C.M., J.T.H., A.P. and K.H. acknowledge funding from the Medical Research Council (MRC) Centre for Global Infectious Disease Analysis (MR/X020258/1) funded by the UK MRC and carried out in the frame of the Global Health EDCTP3 Joint Undertaking supported by the EU; the NIHR for support for the Health Research Protection Unit (HRPU) in Modelling and Health Economics, a partnership between the UK Health Security Agency (UKHSA), Imperial College London, and London School of Hygiene & Tropical Medicine (grant code NIHR200908); E.K. is funded exclusively via the HRPU; T.R., A.P., G.M., G.F. and K.H. acknowledge funding from Community Jameel and Kenneth C Griffin supporting the work of the—Jameel Institute-Kenneth C Griffin Initiative for the Economics of Pandemic Preparedness- at the Jameel Institute, Imperial. C.M. acknowledges support from The Eric and Wendy Schmidt Fund for Strategic Innovation via the Schmidt Polymath Award (G-22-63345). A.P. also acknowledges funding by a joint investigator award to Prof Azra Ghani and KH from the Wellcome Trust (220900/Z/20/Z). The funders of the study had no role in the study design, data collection, data analysis, data interpretation, or writing of the report. For the purpose of open access, the authors have applied a—Creative Commons Attribution' (CC BY) licence to any Author Accepted Manuscript version arising from this submission.

## Author contributions

T.R.: Conceptualization, Methodology, Software, Analysis, Writing. C.M.: Writing, Review and Editing. E.S.K., R.F.: Software, Analysis. J.H., A.P., G.M., G.F.: Review and Editing. A.C.M., M.W.S.: Data Provision. K.H., N.F.: Conceptualization, Methodology, Review and Editing.

## Competing interests

We declare that none of the authors have competing financial or non-financial interests as defined by Nature Portfolio.

## Inclusion and Ethics

All roles and responsibilities were agreed amongst collaborators ahead of the research. Local ethics review was not required due to this being a computational study.
