## [Transparent Peer Review file · Nature Communications]

A mathematical model of H5N1 influenza transmission in US dairy cattle.

Corresponding Author: Dr Thomas Rawson

Version 0:

Reviewer comments:

Reviewer #1

(Remarks to the Author)

The paper presents a meta-population model for H5N1 spread among dairy herds in the United States. Authors use data on cattle movement and current reported incidence to estimate underreporting and expected positivity rates. They also show that interventions to date have had limited impact on dynamics. Finally, they identify sources of uncertainty and states that will require heightened monitoring.

The paper describes a minimal mechanistic model for studying H5N1 transmission across cattle herds. The primary mode of transmission between herds is driven by overall force of infection in the state, while within-herd growth is modeled using SEIR dynamics. Although simplifying the multi-species nature of this outbreak, the model yields various insights into observed epidemiological dynamics and intervention effects, and suggests potential next steps for monitoring and local interventions. However, there are multiple modeling assumptions that may need to be justified or relaxed to make the paper more useful. There could also be a couple of typos in key equations that are worth checking and correcting if needed (I haven't checked if these show up in the code as well). I have listed a few questions/suggestions below, which will need to be addressed before the paper can be considered for acceptance.

Methods: Equation 1:

- The summation term has a denominator that does not depend on j , and the numerator reduces to total number of infected cattle in the state (excluding herd i). In essence, the collective force of infection from all within-state herds is modeled here. This implicitly assumes that there are no heterogeneities in internal trade flows (i.e., herd i has more cattle transfers with herd k)? The authors however mention some notion of locality ('transmission routes existed between neighboring herds') in flows.
- It is worth mentioning that the model assumes no birth-death process, and ignores mortality/culling rate post-outbreak. The latter is low (around 2%), but worth noting in the paper. It is unclear what are the implications of the former on observed dynamics (especially given that lactating cows are at the heart of the outbreak).
- Also, observations suggest not all cattles get infected within a herd (less than 10%). [Pun unintended] Is there 'herd immunity' at play, and is this captured by the model?
- Suggestion 1: Perhaps a simpler form of equation (1) first line would be [in latex notation]: $-\beta S_i \left(\frac{I_i}{N_i} + \alpha \frac{I_{-i}}{N_{-i}} \right)$ where I_{-i} and N_{-i} represent number of infected and total cattle in the state outside herd i .
- Suggestion 2: N^i_{herds} denotes the number of herds within the state. For ease of reading, perhaps a different symbol/notation can be used for it. (N_i represents number of cattle in a particular herd, and N^i_{herds} represents numbers of herds. I am assuming N represents total number of cattle across all herds).
- Equation 2: Should the term S_i show up in the exponent of the first equation since we are modeling draws from population of size S_i already? By first-order Taylor approximation, the current formulation will result in an S_i^2 .
- Another implication of modeling total force of infection from outside herds, is that the stochasticity in inter-herd flows is not explicitly modeled, and everything is absorbed into the overall rate of infection.
- Equation 3 (probability of outbreak)
- This is a custom formulation based on consultation with veterinarians. It is a combination of relative rate of infections and some threshold on absolute number. It would be good to expand on how this was designed (especially choice of the parameters) and how it applies to herds of different sizes.

- is A^{asc} constant across all herds, in all states, estimated based on California? One would imagine there is some variability in the parameterization of Equation 3 across states in terms of proactiveness and detection power. Would be good to justify why the interpretation for underreporting (Figure 2B) is still valid under such conditions.
- Equation 4:
 - Movement of cattle between states is simulated outside Equation 1. It would be good to justify why this is better than building an overall movement/mixing matrix across all herds and refining Equation 1.
 - Since the left hand terms show number of cattle transferred from i to j , shouldn't $M_{\{i,j\}}$ feature on the right hand side?
 - According to the data, are the export flows seasonal in nature? If yes, please justify using a static movement matrix.
- Equation 5: Hypergeometric distribution for the inspection process is a good choice. In reality, does this also trigger a 'detection'/reporting of outbreak for the source herd/state? I didn't see a connection in the manuscript, so unclear if that is done.
- In Table 2, α is described as 'inter-state transmission proportion'. Should this be 'intra-state' or 'inter-herd', since it models transmission between herds within a state? At the risk of being pedantic, it is better to call it as 'coefficient' since it is unclear what it is a proportion of.

Results and Discussion:

- In Figure 2A, why are the 95% CrI for some states (e.g., Kansas, New Mexico, Ohio) much tighter than others?
- For Figure 5, instead of comparing total number of herds/outbreaks, it may be better to show number of states that newly declare outbreaks (or better, importation events) under different testing regimes (since all simulations are seeded in Texas)? This would probably remove the effect of different herd sizes across the states.
- While not currently present, it would have been great to see a modeling treatment of an intra-state and herd-focused intervention and see if those are more impactful. These may help determine the utility of recent USDA investments for voluntary testing programs. A similar treatment for surveillance enhancements (e.g., sentinel testing) may be a value-add to the paper.
- In Discussion, the most notable states for having 'cryptic infections' are listed as Arizona, Wisconsin, Florida. Is this primarily based on having the least 'probability of no outbreaks' conditioned on none reported yet? If so, worth including Indiana (0.083) above Florida (0.094). That said, the interpretations for AZ, WI, and FL are useful context, and is worth presenting in a figure (e.g., x-axis mean herd size, y-axis number of herds, circle size - trade connection with Texas, color - detected/not-detected).
- Tables S1-5 in Supplement could be presented as Figures/Heatmaps to identify and report qualitative patterns (e.g., Table S5 spans many pages, and does not yield to any insights).
- Finally, it is worth discussing the multi-species nature of the outbreak and any limitations due to not modeling the role of spillovers from other species (e.g., poultry, wild birds).

(Remarks on code availability)

Only cursorily went through the codebase. The github.io page is useful and includes vignettes to reproduce plots in the manuscript. The readme in github repo could be more detailed.

Reviewer #2

(Remarks to the Author)

To the authors and editor:

The authors present a stochastic meta-population model for the spread of H5N1 within dairy cattle in the United States, calibrate the model to the available empirical data, and consider counterfactual simulation studies around alternative testing interventions. Overall, I found the model to be well-considered and the manuscript to be well-written. My primary criticism of the manuscript is that there are a few points where further detail would be helpful. My comments below identify these points.

Main Text:

Overall: The way the model is introduced and the results are presented needs significant clarification. The authors state that they are estimating the true size of the current epidemic as well as the under-reporting rates by state. However, based on Figure 2 and Table 1, it appears the model is very poorly calibrated to empirical data, thus putting into question the results of the model. While the performance of the model as compared to reported empirical data is understandable, due to the extremely limited data, this is not well stated by the authors, and would aid the reader tremendously to revise of the main text explaining the assumptions of the model in a very clear manner. Specifically, the final paragraph of the introduction should include some explanation that the model produces expected detections and reported outbreaks based on specific structural assumptions of detection and reporting wholly defined by the sizes of herds and number infected per herd. I strongly recommend some revision to this and other text to clarify when possible.

Additionally, in general the results are written as "Figure X portrays XYZ..." rather than stating or summarizing a result, then supporting it with the figure or table reference. While this is not required, it would improve the readability and interpretation of the results. For example, instead of "Panel 3B and 3C display the mean probability ..." (line 177-178), I would recommend something like "We estimate the mean probability of reporting and outbreak decreases from 41% in CA to X% in ... (Figure 3B)".

Figures and Tables Overall: These should be able to stand alone and be understood with only their captions. For Figure 2, more explanation is needed. Figure 4 needs text indicating these are results based on the simulated epidemic. Figure 5

needs details about the "True measures".

1. Line 123-124 (Page 4) - It would be helpful to the reader to provide an example of a state where this happens.
2. Line 133-134 (Page 4) - It would be helpful to the reader to provide an example of a state where this happens.
3. Figure 2 - The authors have clearly explained why the probability of detection is overestimated for some states due to available data being skewed towards California detection rates (Lines 134-144) and why, in some states, the model mean significantly underestimates the probability of first outbreak detection (Lines 146-150). However, this could be strengthened by a concrete example (e.g., if the statement is interpreted correctly - Utah). Additionally, it may assist the reader to add additional confidence intervals, such as 50% or 80% to narrow the band.
4. The authors consider multiple case ascertainment settings (Lines 167-175) & SI section 3.2.3. However, a paragraph of discussion in the SI of choice to not have state-specific or time-varying ascertainment rates (since testing regimes and ascertainment likely vary significantly both in time and by state) feels warranted, especially since within-state detection rates would seem to be strongly influenced by these choices in addition to the distribution of cattle populations in each state the authors focus on. The authors briefly explain this choice in lines 334-336, but more detail would be useful.
5. Lines 284-286: "Rather, more targeted biosecurity interventions at farm level will be required, and better outreach with industrial partners pursued." Can the authors provide concrete examples of what such interventions might entail?

Methods:

The methods are clearly and concisely presented with sufficient detail to allow replication, particularly when paired with the well-documented software package.

Software:

The software package is well laid out. The vignettes have clear and helpful comments, and the readme guides the user through installation and clearly states the license.

Supplementary Materials:

1. For Table S6, the authors should also provide ESS statistics
2. Some of the larger tables in the SI (for example, table S5, which takes up nine pages) could be provided as supplementary Excel or CSV files instead

(Remarks on code availability)

The software package is well laid out. The vignettes have clear and helpful comments, and the readme guides the user through installation and clearly states the license.

Reviewer #3

(Remarks to the Author)

This manuscript uses stochastic infectious disease dynamics models to estimate the size of the current spread of H5N1 influenza virus among US dairy cattle. The work builds on that model to assess the current systems for surveillance to see whether that system is likely to detect an outbreak once it has occurred and whether that is sufficient to curb the current epidemic. This subject is timely and important, and represents an excellent application of these methods for better understanding the tools we have to combat disease spread among livestock.

The methodology and reasoning in the manuscript are sound. I have a few minor suggestions for improvements but they are largely cosmetic or structural.

Minor suggestions -

I am somewhat unclear on the "interventions" which the authors refer to - how are these implemented in the real world, and how are they implemented in the model itself? Since Figure 5 is intended to demonstrate the impact of those interventions and your main conclusions related to those interventions being insufficient, please elaborate further on what sorts of interventions you included in the three modeled scenarios shown in Figure 5.

One structural suggestion I have is that parts of the methods section appear to be mixed in with reported results. For example, in the part of the manuscript beginning on page 7, as well as Figure 3, you have modeling assumptions described alongside results. I would suggest instead describing the assumptions around ascertainment (Figure 3A) in the methods section, as this enters into your methodology upstream of your modeling results.

One key modeling assumption you include is using the detection probabilities from California - Is the use of California here optimistic or pessimistic? Did you consider instead using detection rates from New Mexico, or even Texas? How would the results change in those instances?

Terrific work!

(Remarks on code availability)

I am not sufficiently trained in C++ to review this code with any detail or accuracy.

Version 1:

Reviewer comments:

Reviewer #1

(Remarks to the Author)

Thanks to the authors for making substantial changes to the manuscript in response to the reviewer comments. Specifically, I appreciate the clarification for the modeling assumptions and improved notations/equations. The results and discussion section have also been improved significantly, providing better context to the evolving nature of the response in US. Finally, I appreciate the new supplementary figures and files that enhance the interpretation of their modeling results.

Overall I commend the authors for undertaking this effort, especially in a limited data setting. The paper also reinforces the use of explainable mechanistic models in making sense of an ongoing public health crisis.

I have no reservations in accepting this manuscript. I look forward to it being published and guiding the next phase of response in the United States.

(Remarks on code availability)

Thanks for sharing the codebase. Very useful for building on this work.

We thank all reviewers for their prompt, thorough, and constructive feedback. In responding to these reviews, we have made substantial changes to the manuscript text to address the points raised by reviewers. This also includes an additional sensitivity analysis and additional results in the supplementary material.

We respond to all points directly below.

REVIEWER COMMENTS

Reviewer #1 (Remarks to the Author):

The paper presents a meta-population model for H5N1 spread among dairy herds in the United States. Authors use data on cattle movement and current reported incidence to estimate underreporting and expected positivity rates. They also show that interventions to date have had limited impact on dynamics. Finally, they identify sources of uncertainty and states that will require heightened monitoring.

The paper describes a minimal mechanistic model for studying H5N1 transmission across cattle herds. The primary mode of transmission between herds is driven by overall force of infection in the state, while within-herd growth is modeled using SEIR dynamics. Although simplifying the multi-species nature of this outbreak, the model yields various insights into observed epidemiological dynamics and intervention effects, and suggests potential next steps for monitoring and local interventions. However, there are multiple modeling assumptions that may need to be justified or relaxed to make the paper more useful. There could also be a couple of typos in key equations that are worth checking and correcting if needed (I haven't checked if these show up in the code as well). I have listed a few questions/suggestions below, which will need to be addressed before the paper can be considered for acceptance.

Methods: Equation 1:

- The summation term has a denominator that does not depend on j , and the numerator

reduces to total number of infected cattle in the state (excluding herd *i*). In essence, the collective force of infection from all within-state herds is modeled here. This implicitly assumes that there are no heterogeneities in internal trade flows (i.e., herd *i* has more cattle transfers with herd *k*)? The authors however mention some notion of locality ('transmission routes existed between neighboring herds') in flows.

We have altered equations (1) as per the useful suggestions below, which we agree are now easier to interpret. This does, indeed, mean that there are no differences in force of infection between herds within the same state due to a lack of informing data on the extent to which shared materials/personnel/environmental runoff link herds. This is a model assumption that has been made more clear in section 2.1, page 4, of the SI, with the additional lines: "*We note that, under this modeling assumption, there is no difference in the force of infection one herd applies to all other herds in the US state it resides in. Further spatial kernel approaches would require substantial additional information on the precise geo-location of each dairy herd, and the extent to which equipment, workforce, and services is shared between specific herds*".

This is however different to the actual movement of cattle between herds within the same state. This is conducted via the same mechanism as for inter-state movements of cattle (though not subject to potential cancellation following border testing). Heterogeneities in internal trade flows exist due to the size of prospective destination herds scaling their respective probabilities of being chosen as the destination herd. Lines 460-463 (unchanged) state: "*Once the destination state is determined, we randomly allocate which herd in the destination state the cattle will be exported to, scaled by the population size of the respective herds, to preserve herd sizes*."

We agree that some confusion may come from our previous wording of "*between neighboring herds*", which implicitly suggests some degree of spatial factoring. This is not the case. We have altered this wording to be clearer, and align more with the related citation [37], which speaks to the transmission risks posed by shared equipment/staff or wild birds, and does not specifically implicate herds which are geographically adjacent to each other. The sentence now reads, on lines 423-427; "*Early epidemiological surveys of farms reporting outbreaks found that transmission routes existed between herds in the same state through the shared use of equipment, staff, or the movements of wild birds [37], which we capture here in the model*".

- It is worth mentioning that the model assumes no birth-death process, and ignores mortality/culling rate post-outbreak. The latter is low (around 2%), but worth noting in the paper. It is unclear what are the implications of the former on observed dynamics (especially given that lactating cows are at the heart of the outbreak).

Yes, this warrants being made clear. We chose not to include birth/death processes due to the relatively short length of the study period. Additionally, the 2% culling figure links only to identified and reported outbreaks. To explore this process, we have additionally implemented the option for birth/death processes (of equal flux) within the cowflu package (see PRs 33-35 for code specifics: <https://github.com/mrc-ide/cowflu/pull/33>) and added a new sensitivity analysis to the SI exploring their dynamic impact. SI section 3.2.4 shows simulations including these processes with a death rate of $\mu = 1/305$ weeks (corresponding to an average dairy cow lifespan of 5.87 years as cited in SI). We see that their inclusion has insignificant impact on the model results. We have made this explicitly clear now via lines 249-252 of the main manuscript: "*Due to the relatively short time frame considered, and unclear evidence as to the extent of mortality or culling, we did not include birth-death*

processes within our model. Supplementary Material section 3.2.4 considers the dynamic impact of including such birth-death mechanisms.”. We cite the “2% or less” value in the related SI section.

- Also, observations suggest not all cattles get infected within a herd (less than 10%). [Pun unintended] Is there ‘herd immunity’ at play, and is this captured by the model?

Herd immunity effects are captured in so far as standard SEIR dynamics will reduce the force of infection as the susceptible proportion of the herd depletes. We do not mechanistically enforce a custom carrying capacity of infected cattle. While morbidity was reported as <10% in early epidemiological reports (<https://www.aphis.usda.gov/sites/default/files/hpai-dairy-national-epi-brief.pdf>), virological studies such as the cited Caserta et al. (2024) were able to detect viral RNA in non-clinical animals as well (<https://www.nature.com/articles/s41586-024-07849-4>). Accurate assessment of herd-level proliferation would require substantial sentinel testing that does not yet exist. We have made these points clearer in the in-depth methods section of the Supplementary Material, section 2.1, page 4, through the additional lines: “While early epidemiological reports highlighted that < 10% of cows on infected herds displayed clinical symptoms [2], virological surveys of these farms identified viral H5N1 RNA within non-clinical animals on infected premises as well [3]. Due to this uncertainty, we do not enforce strict infection carrying capacities into the within-herd model dynamics.”.

- Suggestion 1: Perhaps a simpler form of equation (1) first line would be [in latex notation]: $-\beta S_i \left(\frac{I_i}{N_i} + \alpha \frac{I_{-i}}{N_{-i}} \right)$ where I_{-i} and N_{-i} represent number of infected and total cattle in the state outside herd i .

We agree this is a far more interpretable format and have adjusted equations (1) accordingly.

- Suggestion 2: N^i_{herds} denotes the number of herds within the state. For ease of reading, perhaps a different symbol/notation can be used for it. (N_i represents number of cattle in a particular herd, and N^i_{herds} represents numbers of herds. I am assuming N represents total number of cattle across all herds).

We agree that the inclusion of the N^i_{herds} term was not clear. By implementing suggestion 1 above, the term no longer features in the equations.

- Equation 2: Should the term S_i show up in the exponent of the first equation since we are modeling draws from population of size S_i already? By first-order Taylor approximation, the current formulation will result in an S_i^2 .

Thank you for spotting this, this was indeed simply a typo and does not appear in the codebase (see lines 225/226 of model source code). We have corrected equations (2) in the manuscript and SI.

- Another implication of modeling total force of infection from outside herds, is that the stochasticity in inter-herd flows is not explicitly modeled, and everything is absorbed into the overall rate of infection.

See previous response to Methods: Equation 1 RE: inter-herd flow assumptions.

- Equation 3 (probability of outbreak)

- This is a custom formulation based on consultation with veterinarians. It is a combination of relative rate of infections and some threshold on absolute number. It would be good to expand on how this was designed (especially choice of the parameters) and how it applies to herds of different sizes.

We have expanded section 2.3, page 6, of the SI ("Reporting an outbreak"), considerably to better explain the motivation behind these assumptions, with examples of different herd sizes. The following paragraph has been added:

"These modeling assumptions were developed in consultation with veterinarian staff, informed by their experiences of the stage at which illnesses in a herd are reported. Assuming a number of cows in a herd are stricken by behavioral changes and reduced milk production, how many cows must exhibit such symptoms before a veterinarian is consulted? Starting from experiences with large holdings of thousands of dairy cattle, the point at which an infection influencing milk production is more likely to be noticed than not, was considered to be approximately 100 infected cattle. At this stage, not only would a sufficiently large number of cattle be identified within the same milking line, but a communicable pathogen is likely to be suspected. Hence, the yellow "50% probability of ascertainment" region of Figure 3A in the main manuscript, is seen to intercept the x-axis at 100 infected cows. Naturally, however, for smaller population herds, an infection passes the point of "more likely to be identified than not" at a smaller number of infected cattle. This point scales with herd population size to a limit such as, in a herd of only 2 cattle, an infection is more likely to be identified than not, as soon as 1 of the 2 cows is infected. Hence, the yellow "50% probability of ascertainment" region of Figure 3A in the main manuscript, is seen to intercept the y-axis at 50% of the herd infected. Note that these values are then further scaled by the fit model parameter A^{asc} ."

- is A^{asc} constant across all herds, in all states, estimated based on California? One would imagine there is some variability in the parameterization of Equation 3 across states in terms of proactiveness and detection power. Would be good to justify why the interpretation for underreporting (Figure 2B) is still valid under such conditions.

A^{asc} is a fit model parameter, that indeed holds across all herds, in all states. There are undoubtedly differences in rates of detection between the US states. The choice then is between fitting one global parameter (as we have done), or individual A^{asc} parameters for each state. The issue with the latter approach is that, since many US states have reported 0 outbreaks to date, their individual A^{asc} parameters would, naturally, fit tightly to 0 to maximise likelihood. This would severely weaken the usefulness of the model as, two states which have both reported 0 outbreaks – Arizona and Rhode Island, would both fit respective A^{asc} close to 0, and would see very close fits to the data (or lack thereof) in Figure 2B. Whereas, by opting for one global parameter, we are able to observe that, assuming all states report outbreaks equally, it is not surprising that we see no outbreaks in Rhode Island, based on the underlying cattle trade dynamics. It is however surprising that we see no outbreaks in Arizona, thus suggesting that under-reporting is to blame.

In short, there is simply not enough outbreaks data to adequately identify state-specific A^{asc} parameters.

We have made this clearer in the Methods section, lines 443-447: "While US states undoubtedly vary in their detection capabilities, there is insufficient outbreak data to fit

unique A^{asc} values for each state. Assuming one national A^{asc} parameter allows us to identify which states that have reported 0 outbreaks to date are driven mostly by under-reporting (Figure 2B).” and in greater detail in SI section 2.3 “Reporting an outbreak”, page 6:

“While different US states are undoubtedly likely to differ in their rates of detection subject to public health staffing or resourcing, fitting individual state level ascertainment scalings would require some degree of observations in all states. The majority of US states have not declared any outbreaks to date. If we were to fit unique A^{asc} values for each state, these fit values would generate posterior distributions very close to 0, to maximise the likelihood function. By assuming one global parameter, we are instead able to identify which states it is reasonable to have observed no outbreaks in, and which it is unreasonable - suggesting under-reporting. For example, Arizona and Rhode Island have both yet to declare any outbreaks of H5N1 in dairy cattle within the state. Our model results in Figure 2B of the main manuscript, however, demonstrate that while that is a very plausible outcome in the case of Rhode Island, that is unlikely to be a true representation of the disease burden in Arizona.”

- Equation 4:

- Movement of cattle between states is simulated outside Equation 1. It would be good to justify why this is better than building an overall movement/mixing matrix across all herds and refining Equation 1.

Due to the multiple stochastic steps involved in co-ordinating the movement of cattle, correctly including them within equations (1) would require first defining the matrix $M_{i,j}$, then the vectors p_{export} and p_{export_size} , then defining mathematically an associated 4-dimensional binary matrix $M^{i,j}_{k,l}$ detailing the destination herd $\{k,l\}$ ({herd , state}) for each origin herd $\{i,j\}$, unique to each time step dt , and a time-dependent variable depicting the hypergeometric draws capturing border testing. The amount of notation this would add to equations (1) quickly becomes very large. Given the inter-disciplinary readership of Nature Communications, we felt it more approachable to provide the underlying epidemiological dynamics as equations, and dedicate more text and explanation to the additional stochastic steps.

- Since the left hand terms show number of cattle transferred from i to j , shouldn't $M_{i,j}$ feature on the right hand side?

The number of cattle being exported does not depend on the destination herd/state, only on the population of the exporting herd. The series of steps determining the movement of cattle,

- 1) Will this herd export cattle?
- 2) If so, which US state will it export them to?
- 3) Which herd in that destination state will it export them to?
- 4) How many cattle (from which epidemiological compartments) will it send?
- 5) Will infected cattle be detected if crossing state borders?

are described in detail in the pseudocode of section 2.2 of the SI, and the subsequent SI sections. One could collapse steps (1),(2) and (3) above by then including the stochastic draws related to matrix $M_{i,j}$ and p_{export} within equations (4), but at the cost of the considerably increased mathematical notation referenced in the previous comment. We appreciate that the confusion was likely introduced from our using indices i and j initially to depict herds, and then using them additionally for US states. We now use k and l indices for US states in the main manuscript to help delineate.

- According to the data, are the export flows seasonal in nature? If yes, please justify using a static movement matrix.

Investigating the USAMM movement data demonstrates only minor, non-significant seasonal variation, hence justifying this fixed value assumption. We have added a new plot demonstrating this to the SI (Figure S3), noting in the accompanying text that the mean value never falls outside the max/min range of the 95% CrI for the entire time series.

- Equation 5: Hypergeometric distribution for the inspection process is a good choice. In reality, does this also trigger a 'detection'/reporting of outbreak for the source herd/state? I didn't see a connection in the manuscript, so unclear if that is done.

We agree this was unclear as written and have added lines 473-474 to state this explicitly: *"Note, a positive test prevents the export, but does not immediately register as a reported outbreak."* Indeed, one of the main actions we stress in our conclusion is to make this testing data publicly available. Currently, a positive test in preparation for state border crossing does not automatically necessitate a "reporting". Farmers simply must demonstrate a negative test to carry out an export, it is currently not clear how often a positive test result would eventually be reported to the USDA or CDC.

- In Table 2, α is described as 'inter-state transmission proportion'. Should this be 'intra-state' or 'inter-herd', since it models transmission between herds within a state? At the risk of being pedantic, it is better to call it as 'coefficient' since it is unclear what it is a proportion of.

You are correct, this should be "intra-state" and was a typo on our behalf. Thank you for spotting this. It has been corrected and "proportion" replaced with "coefficient".

Results and Discussion:

- In Figure 2A, why are the 95% CrI for some states (e.g., Kansas, New Mexico, Ohio) much tighter than others?

As written in the methods section lines 410-411: “we also seeded 9 additional herds in accordance with the nine early outbreaks detailed in Caserta et al. (2024)”. The 9 herds detailed in Caserta et al. (2024)’s paper are located in Texas, Ohio, New Mexico, and Kansas, and a large absolute number of infected cows were reported in the manuscript, meaning they are quickly identified as outbreaks. Hence, there is a very thin credible interval for the “Time of First Outbreak Detection” in these states. We have mentioned this earlier now in the results section to explain this feature, far sooner than the Methods section. Lines 168-170 read: “Particularly narrow 95% CrIs are seen in Figure 2A for Texas, Ohio, New Mexico, and Kansas, due to the seeding of cases in these states as detailed in the Methods.”.

- For Figure 5, instead of comparing total number of herds/outbreaks, it may be better to show number of states that newly declare outbreaks (or better, importation events) under different testing regimes (since all simulations are seeded in Texas)? This would probably remove the effect of different herd sizes across the states.

Due to the wide uncertainty in “time of first outbreak detected” (Figure 2A), altering Figure 5 to instead show the number of states reporting an outbreak would equally have considerably broad 95% CrIs. The majority of the reporting and messaging around the ongoing epidemic in dairy cattle refers to “outbreaks detected” or “infected herds”, hence our decision to prioritise this as the measure by which to compare the impact of the border testing procedures.

- While not currently present, it would have been great to see a modeling treatment of a intra-state and herd-focused intervention and see if those are more impactful. These may help determine the utility of recent USDA investments for voluntary testing programs. A similar treatment for surveillance enhancements (e.g., sentinel testing) may be a value-add to the paper.

As an additional result, we have run a counterfactual scenario whereby we consider the potential impact through herd-focused interventions. We mechanistically capture this by re-running the baseline analysis but halving the posterior draws of the alpha parameter, thus considering reduced transmission of disease between herds in the same US state. This could be achieved through measures such as disinfecting of equipment between sites, workers changing clothes and washing their hands between sites, or quarantining suspected infected cattle from outside grazing. The counterfactual shows a more significant impact in reducing disease burden than stricter border testing, however it is still incapable of outright stopping the spread of disease. This additional result is added as Figure S19 in section 3.1 “Additional Results”, page 23, of the Supplementary Material. In addressing later reviewer comments in precisely suggesting potential interventions, we refer to this new result throughout the main manuscript. (see later responses for related line numbers).

- In Discussion, the most notable states for having 'cryptic infections' are listed as Arizona, Wisconsin, Florida. Is this primarily based on having the least 'probability of no outbreaks' conditioned on none reported yet? If so, worth including Indiana (0.083) above Florida (0.094). That said, the interpretations for AZ, WI, and FL are useful context, and is worth

presenting in a figure (e.g., x-axis mean herd size, y-axis number of herds, circle size - trade connection with Texas, color - detected/not-detected).

These states were highlighted not just for their low “probability of no outbreaks” values, but also the large size of their dairy industries. Indiana fulfils both these criteria, and should certainly have been included in this list. This has now been added, see line 268, and lines 275 to 277: “Indiana presents itself as having a high likelihood of probable infection due both to having a very high number of dairy herds, but also due to its frequent trading links with Wisconsin.”

We have produced the figure you suggested (see below) and included this as Figure S20 within SI section 3.1 “Additional Results”. We refer to it in the main manuscript at lines 281-284: “Figure S20 of the Supplementary Material visualises the herd population sizes of each state against the frequency of imports from Texas, demonstrating the relationship between herd sizes and outbreak likelihood.”

- Tables S1-5 in Supplement could be presented as Figures/Heatmaps to identify and report qualitative patterns (e.g., Table S5 spans many pages, and does not yield to any insights).

Table S1 can now be represented by the above scatter plot. As per your suggestion, we have added the following three heatmaps to the Supplementary Material, figures S2, S4, and S5. The previous multi-page table has been moved to an additional supplementary .csv file as per Reviewer 2’s suggestion.

Weekly Probability of Exporting Cattle per Herd, by US State

Mean Proportion of Herd Exported by US State

Heatmap of Movement Matrix Values

- Finally, it is worth discussing the multi-species nature of the outbreak and any limitations due to not modeling the role of spillovers from other species (e.g., poultry, wild birds).

We have added a paragraph to the Discussion raising this point, at lines 388-395:
“Additionally, our work does not consider the dynamic impact of other zoonotic reservoirs. The ongoing H5N1 epidemic in the US is also heavily impacting the poultry industry, with 662 counties reporting outbreaks as of March 3rd 2025 [34]. Modeling the disease in poultry is significantly more challenging due to the role played by wild bird migration [35], and our current model does not consider spillover from other animal populations. Further work identifying farm sites which house multiple host species would be an important next step in identifying points of spillover risk between reservoir animals, presenting a risk of further genetic reassortment.”

Reviewer #1 (Remarks on code availability):

Only cursorily went through the codebase. The github.io page is useful and includes vignettes to reproduce plots in the manuscript. The readme in github repo could be more detailed.

Thank you, we have added additional information to the package README.

Reviewer #2 (Remarks to the Author):

To the authors and editor:

The authors present a stochastic meta-population model for the spread of H5N1 within dairy cattle in the United States, calibrate the model to the available empirical data, and consider counterfactual simulation studies around alternative testing interventions. Overall, I found the model to be well-considered and the manuscript to be well-written. My primary criticism of the manuscript is that there are a few points where further detail would be helpful. My comments below identify these points.

Main Text:

Overall: The way the model is introduced and the results are presented needs significant clarification. The authors state that they are estimating the true size of the current epidemic as well as the under-reporting rates by state. However, based on Figure 2 and Table 1, it appears the model is very poorly calibrated to empirical data, thus putting into question the results of the model. While the performance of the model as compared to reported empirical data is understandable, due to the extremely limited data, this is not well stated by the authors, and would aid the reader tremendously to revise of the main text explaining the

assumptions of the model in a very clear manner. Specifically, the final paragraph of the introduction should include some explanation that the model produces expected detections and reported outbreaks based on specific structural assumptions of detection and reporting wholly defined by the sizes of herds and number infected per herd. I strongly recommend some revision to this and other text to clarify when possible.

We have added this specific addition to the Introduction to better address these modelling assumptions from the outset. Lines 93-100 now read: *“Mechanistic modeling assumptions are made relating the probability of detecting and reporting an infected herd proportional to the number of infected cattle and total population size of the herd, irrespective of the US state they reside in. The model successfully simulates outbreaks for US states that have frequently reported outbreaks, such as California. We estimate the rates of under-reporting by state, by comparing the number of confirmed outbreaks with model simulated trajectories, and present the anticipated rates of positivity for cattle tested upon leaving each state over time.”*.

Our existing explanation of these features was also buried within a larger paragraph, we have separated these now to make our discussion on model performance compared to data easier to identify. Lines 142-152 read: *“The model is seen to overestimate the number of outbreaks in some states. While our model assumes differences in outbreak detection due to differences in herd sizes by state, we do not assume further intrinsic state-varying differences in outbreak detection. In reality, differences in public health resourcing and messaging will impact outbreak detection rates. 72% of outbreaks reported as of December 9th 2024 have been in California. Due to making up the majority of the epidemiological data, model fits are mostly tuned to the detection rates observed in California. Therefore, overestimation of the model can be interpreted as under-reporting within a state compared broadly to baseline reporting efforts in California.”*.

Additionally, in general the results are written as “Figure X portrays XYZ...” rather than stating or summarizing a result, then supporting it with the figure or table reference. While this is not required, it would improve the readability and interpretation of the results. For example, instead of “Panel 3B and 3C display the mean probability ...” (line 177-178), I would recommend something like “We estimate the mean probability of reporting and outbreak decreases from 41% in CA to X% in ... (Figure 3B)”.

We have re-formatted in this style in multiple places throughout the manuscript, and agree that this improves interpretability. This includes your specific example. Lines 197-203 now read *“We calculate the mean probability that a randomly selected herd in each state will report an outbreak, given that 10% of its animals are infected. These values ranged from 0.412 in California, a states with a greater number of large herds, to 0.092 in West Virginia (Figures 3B and 3C). We see that states with a greater number of large herds are more likely to report outbreaks than other states. Correspondingly, California has reported the vast majority of outbreaks to date (Table 1).”*,

Additionally, lines 156-162 when discussing Table 1 now read: *“26 of the 48 US states (54%) observed an outbreak of H5N1 before December 2nd 2024 in the majority of model simulations (> 50% of simulations, Table 1). Based on these probabilities, one would expect to have observed outbreaks in a mean of 27 (22-32 95% CrI) states by December 2nd 2024, assuming all states reported outbreaks equally. In actuality, only 16 states identified and reported outbreaks in this time period, indicating a high degree of under-reporting compared to the high baseline set by California.”*.

Other such changes are made throughout the text. However, in some instances, the complexity or aim of the figure (especially in the case of the schematic Figure 1) made such a change difficult. In some of these rare instances we have kept the original wording where we felt it afforded better interpretability.

Figures and Tables Overall: These should be able to stand alone and be understood with only their captions. For Figure 2, more explanation is needed. Figure 4 needs text indicating these are results based on the simulated epidemic. Figure 5 needs details about the “True measures”.

We have expanded the Figure captions as you suggested. The figure 2 caption now reads:

“Model Simulations. After fitting model parameters we simulate 20,000 stochastic realizations drawing from the parameter posterior distributions. Displayed is the epidemic trajectory from these simulations for each US state. **(A)** shows the date at which the first outbreak is detected in a state, a binary outcome. 0 indicates the state has not yet reported its first outbreak. 1 indicates that it has. Model simulation thus plots the proportion of the 20,000 realizations which have simulated a reported outbreak by this date. **(B)** shows the proportion of herds in each state which report new outbreaks per week, assuming no differences in ascertainment (parameter A^{asc}) between states. Red points depict data. The black line depicts the model mean, the shaded grey region depicts the 95% credible interval (95% CrI).”

The figure 4 caption now reads:

“Probability of positive border testing. We calculate the probability of an export of cattle out of each state testing positive from 20,000 stochastic model simulations. When moving cattle inter-state, up to 30 cattle will be tested for H5N1 per export. Panels show the state average per-herd probability that, should a herd export cattle, it would test positive at: **(A)** week beginning April 15th 2024, **(B)** week beginning August 19th 2024, and **(C)** week beginning December 2nd 2024.”

The figure 5 caption now reads:

“Testing intervention counterfactuals. **(A)** The number of new reported outbreaks weekly. **(B)** The number of herds nationally with any infected cattle. **(C)** The total number of infected cows nationally over time. Solid lines show simulation mean. Shaded regions show 95% CrI. Blue (“True measures”) depicts baseline model assumptions, whereby up to 30 cows in each inter-state export are tested starting from April 29th 2024. Red depicts the scenario with no border testing. Green depicts border testing of up to 100 cows from each export, implemented 28 days earlier, on April 1st 2024.”

1. Line 123-124 (Page 4) - It would be helpful to the reader to provide an example of a state where this happens.

We have added “... ,such as Washington,...” to better direct the reader to the phenomenon we describe.

2. Line 133-134 (Page 4) - It would be helpful to the reader to provide an example of a state where this happens.

We have added the line “For example, Texas, New Mexico, and Ohio all feature simulations whose credible interval does not contain the observed data.”

3. Figure 2 - The authors have clearly explained why the probability of detection is overestimated for some states due to available data being skewed towards California detection rates (Lines 134-144) and why, in some states, the model mean significantly underestimates the probability of first outbreak detection (Lines 146-150). However, this could be strengthened by a concrete example (e.g., if the statement is interpreted correctly - Utah). Additionally, it may assist the reader to add additional confidence intervals, such as 50% or 80% to narrow the band.

In addition to the previous precise state examples, we also recreated Figure 2 including 50% Crls as per your suggestion (see below).

A Date of First Outbreak Detection

B Proportion of Herds Declaring Outbreaks Weekly

We have also given a specific example as per your suggestion, highlighting Arizona as a particularly clear example of suspected under-reporting. Lines 150-152 read: *“Therefore, overestimation of the model can be interpreted as under-reporting within a state compared broadly to baseline reporting efforts in California, as seen most strongly in the case of Arizona (Figure 2A)”*.

4. The authors consider multiple case ascertainment settings (Lines 167-175) & SI section 3.2.3. However, a paragraph of discussion in the SI of choice to not have state-specific or time-varying ascertainment rates (since testing regimes and ascertainment likely vary significantly both in time and by state) feels warranted, especially since within-state detection rates would seem to be strongly influenced by these choices in addition to the distribution of cattle populations in each state the authors focus on. The authors briefly explain this choice in lines 334-336, but more detail would be useful.

See our previous response to Reviewer 1 on this same query. We have now made this clearer in the Methods section, lines 443-447: *“While US states undoubtedly vary in their detection capabilities, there is insufficient outbreak data to fit unique A^{asc} values for each state. Assuming one national A^{asc} parameter allows us to identify which states that have reported 0 outbreaks to date are driven mostly by under-reporting (Figure 2B).”* and in greater detail in SI section 2.3 “Reporting an outbreak”, page 6:

“While different US states are undoubtedly likely to differ in their rates of detection subject to public health staffing or resourcing, fitting individual state level ascertainment scalings would require some degree of observations in all states. The majority of US states have not declared any outbreaks to date. If we were to fit unique A^{asc} values for each state, these fit values would generate posterior distributions very close to 0, to maximise the likelihood function. By assuming one global parameter, we are instead able to identify which states it is reasonable to have observed no outbreaks in, and which it is unreasonable - suggesting under-reporting. For example, Arizona and Rhode Island have both yet to declare any outbreaks of H5N1 in dairy cattle within the state. Our model results in Figure 2B of the main manuscript, however, demonstrate that while that is a very plausible outcome in the case of Rhode Island, that is unlikely to be a true representation of the disease burden in Arizona.”

5. Lines 284-286: “Rather, more targeted biosecurity interventions at farm level will be required, and better outreach with industrial partners pursued.” Can the authors provide concrete examples of what such interventions might entail?

Yes, we have given more precise examples including relevant citations supporting these measures. Lines 315-320 read: *“This suggests that targeted biosecurity interventions at farm level, such as postmilking teat dipping and the use of disposable wipes for premilking teat disinfection [25], and interventions between herds such as boot dips at facility entrances, clothing disinfection post-site visit, or greater emphasis on adequate personal protective equipment (PPE) [26] will be required (Supplementary Figure S19)”*.

Methods:

The methods are clearly and concisely presented with sufficient detail to allow replication, particularly when paired with the well-documented software package.

Software:

The software package is well laid out. The vignettes have clear and helpful comments, and the readme guides the user through installation and clearly states the license.

Supplementary Materials:

1. For Table S6, the authors should also provide ESS statistics

We have added ESS to the table.

2. Some of the larger tables in the SI (for example, table S5, which takes up nine pages) could be provided as supplementary Excel or CSV files instead

We have removed Table S5 from the SI as per your suggestion and instead included it as a separate supplementary CSV item.

Reviewer #2 (Remarks on code availability):

The software package is well laid out. The vignettes have clear and helpful comments, and the readme guides the user through installation and clearly states the license.

Reviewer #3 (Remarks to the Author):

This manuscript uses stochastic infectious disease dynamics models to estimate the size of the current spread of H5N1 influenza virus among US dairy cattle. The work builds on that model to assess the current systems for surveillance to see whether that system is likely to detect an outbreak once it has occurred and whether that is sufficient to curb the current epidemic. This subject is timely and important, and represents an excellent application of these methods for better understanding the tools we have to combat disease spread among livestock.

The methodology and reasoning in the manuscript are sound. I have a few minor suggestions for improvements but they are largely cosmetic or structural.

Minor suggestions -

I am somewhat unclear on the "interventions" which the authors refer to - how are these implemented in the real world, and how are they implemented in the model itself? Since Figure 5 is intended to demonstrate the impact of those interventions and your main conclusions related to those interventions being insufficient, please elaborate further on what sorts of interventions you included in the three modeled scenarios shown in Figure 5.

The interventions considered in Figure 5 consider alternate variations on the only enforced interventions to date – namely testing of cattle crossing from one state to another. The expanded figure caption following Reviewer 2's suggestion makes these counterfactuals clearer. Additionally, lines 222-227 read: *"Lastly, we use the model to assess the impact that interstate testing has had on the epidemic trajectory. We consider two counterfactual scenarios. Scenario 1) weaker measures - we assume no restrictions are introduced, no testing is required when exporting cattle, and thus all interstate exports proceed unabated. Scenario 2) stronger measures - we assume that the federal order was implemented 28 days earlier, on April 1st 2024, and that up to 100 cattle are tested instead of 30."*

One structural suggestion I have is that parts of the methods section appear to be mixed in with reported results. For example, in the part of the manuscript beginning on page 7, as well as Figure 3, you have modeling assumptions described alongside results. I would suggest instead describing the assumptions around ascertainment (Figure 3A) in the methods section, as this enters into your methodology upstream of your modeling results.

Thank you for this suggestion. In response to Reviewer 2 we have actually expanded the degree of modelling assumption explanations provided earlier in the text, including in the Introduction; lines 93-100 now read: *“Mechanistic modeling assumptions are made relating the probability of detecting and reporting an infected herd proportional to the number of infected cattle and total population size of the herd, irrespective of the US state they reside in. The model successfully simulates outbreaks for US states that have frequently reported outbreaks, such as California. We estimate the rates of under-reporting by state, by comparing the number of confirmed outbreaks with model simulated trajectories, and present the anticipated rates of positivity for cattle tested upon leaving each state over time.”*.

We hope this and the additional methodological explanations provided in response to reviewer comments help smooth the interpretability; providing the modelling assumptions needed to fully interpret the results, while also leaving the drier methodological details to the final Methods section, as per the Nature Communications style guidance. Our initial reasoning for including the ascertainment details earlier in the manuscript was due to the fact that these assumptions are important to the interpretation of results, helping to explain why some states observe outbreaks differently despite a global ascertainment scaling parameter. It is arguably a theoretical result, demonstrating how the specific meta-population structure has impacted case ascertainment during this outbreak.

One key modeling assumption you include is using the detection probabilities from California - Is the use of California here optimistic or pessimistic? Did you consider instead using detection rates from New Mexico, or even Texas? How would the results change in those instances?

Thank you, many of our new additions in response to Reviewers 1 and 2 address these assumptions around detection probabilities in greater detail. California, by all accounts, appears to have been effective in reporting outbreaks. Indeed, as of today, 754 outbreaks have been reported in California - the majority of the 1,117 herds in the state. To consider “using” detection rates from other states would require either removing California from the model fitting process, or artificially scaling the global ascertainment parameter, A^{asc} , to better align with the observed outbreak data in other states. Doing this would, naturally, reduce the anticipated number of outbreaks across the country, but would instead mark California as a significant outlier. Under-reporting can be interpreted as a lack of public health resourcing. Over-reporting would not be so easily explained!

Echoing our previous additions in response to Reviewer 1, we have discussed A^{asc} assumptions in the Methods section, lines 443-447: *“While US states undoubtedly vary in their detection capabilities, there is insufficient outbreak data to fit unique A^{asc} values for each state. Assuming one national A^{asc} parameter allows us to identify which states that have reported 0 outbreaks to date are driven mostly by under-reporting (Figure 2B).”* and in greater detail in SI section 2.3 “Reporting an outbreak”, page 6:

“While different US states are undoubtedly likely to differ in their rates of detection subject to public health staffing or resourcing, fitting individual state level ascertainment scalings would require some degree of observations in all states. The majority of US states have not declared any outbreaks to date. If we were to fit unique A^{asc} values for each state, these fit values would generate posterior distributions very close to 0, to maximise the likelihood function. By assuming one global parameter, we are instead able to identify which states it is reasonable to have observed no outbreaks in, and which it is unreasonable - suggesting under-reporting. For example, Arizona and Rhode Island have both yet to declare any outbreaks of H5N1 in dairy cattle within the state. Our model results in Figure 2B of the main manuscript, however, demonstrate that while that is a very plausible outcome in the case of Rhode Island, that is unlikely to be a true representation of the disease burden in Arizona.”

Terrific work!

Reviewer #3 (Remarks on code availability):

I am not sufficiently trained in C++ to review this code with any detail or accuracy.